# Verifying the Union of Manifolds Hypothesis for Image Data

**Bradley C.A. Brown**[*]
University of Waterloo
bcabrown@uwaterloo.ca

**Anthony L. Caterini**
Layer 6 AI
anthony@layer6.ai

**Brendan Leigh Ross**
Layer 6 AI
brendan@layer6.ai

**Jesse C. Cresswell**
Layer 6 AI
jesse@layer6.ai

**Gabriel Loaiza-Ganem**
Layer 6 AI
gabriel@layer6.ai

## Abstract

Deep learning has had tremendous success at learning low-dimensional representations of high-dimensional data. This success would be impossible if there was no hidden low-dimensional structure in data of interest; this existence is posited by the *manifold hypothesis*, which states that the data lies on an unknown manifold of low intrinsic dimension. In this paper, we argue that this hypothesis does not properly capture the low-dimensional structure typically present in image data. Assuming that data lies on a single manifold implies intrinsic dimension is identical across the entire data space, and does not allow for subregions of this space to have a different number of factors of variation. To address this deficiency, we consider the *union of manifolds hypothesis*, which states that data lies on a disjoint union of manifolds of varying intrinsic dimensions. We empirically verify this hypothesis on commonly-used image datasets, finding that indeed, observed data lies on a disconnected set and that intrinsic dimension is not constant. We also provide insights into the implications of the union of manifolds hypothesis in deep learning, both supervised and unsupervised, showing that designing models with an inductive bias for this structure improves performance across classification and generative modelling tasks. Our code is available at https://github.com/layer6ai-labs/UoMH.

## 1 Introduction

The manifold hypothesis (Bengio et al., 2013) states that high-dimensional data of interest often lives in an unknown lower-dimensional manifold embedded in ambient space, and there is strong evidence supporting this hypothesis. From a theoretical perspective, it is known that both manifold learning and density estimation scale exponentially with the (low) *intrinsic* dimension when such structure exists (Ozakin & Gray, 2009; Narayanan & Mitter, 2010), while scaling exponentially with the (high) *ambient* dimension otherwise (Cacoullos, 1966). Thus, the most plausible explanation for the success of machine learning methods on high-dimensional data is the existence of far lower intrinsic dimension, which facilitates learning on datasets of fairly reasonable size. This is verified empirically by Pope et al. (2021), in which a comprehensive study estimating the intrinsic dimension of commonly-used image datasets is performed, clearly finding low-dimensional structure.

However, thinking of observed data as lying on a single unknown low-dimensional manifold is quite limiting, as this implies that the intrinsic dimension throughout the dataset is constant. If we consider the intrinsic dimensionality to be the number of factors of variation generating the data, we can see that this formulation prevents distinct regions of the data's support from having differing quantities of factors of variation. Yet this seems to be unrealistic: for example, we should not expect the number of factors needed to describe 8s and 1s in the MNIST dataset (LeCun et al., 1998) to be equal.

To accommodate this intuition, in this paper we consider the *union of manifolds* hypothesis: that high-dimensional image data often lies not on a single manifold, but on a disjoint union of manifolds

---

[*]Work done during an internship at Layer 6 AI.

of *different intrinsic dimensions*.[1] While this hypothesis has motivated work in the clustering literature (Vidal, 2011; Elhamifar & Vidal, 2011; 2012; 2013; Zhang et al., 2019; Abdolali & Gillis, 2021; Cai et al., 2022), to the best of our knowledge it has never been empirically explored analogously to the way that Pope et al. (2021) probe the manifold hypothesis. In this work we carry out this verification on commonly-used image datasets, first by confirming their supports are disconnected, and then by estimating the intrinsic dimension on each component, finding that there is indeed variation in these estimates. In order to verify that the support of the data is disconnected, we leverage *pushforward deep generative models* (Salmona et al., 2022; Ross et al., 2022), which generate samples by transforming noise through a neural network $G$. We first prove that these deep generative models (DGMs) are incapable of modelling disconnected supports. We then argue that the class labels provided in our considered datasets approximately identify connected components (i.e. different classes are mostly disconnected from each other), and show that training a pushforward model on each class outperforms training a single such model on the entire dataset, even when using the same computational budget: this improvement is a firm indicator that the support is truly disconnected.

After empirically verifying the union of manifolds hypothesis, we turn our attention to some of its implications in deep learning. We establish that classes with higher intrinsic dimension are harder to classify, and guided by this insight, we also show that classification accuracy can be improved by more heavily weighting the terms corresponding to classes of higher intrinsic dimension in the cross entropy loss. We also show that the same DGMs we used to confirm the disconnectedness of the data support, which we call disconnected DGMs, provide a performant class of models which are competitive with their non-disconnected baselines across a wide range of datasets and models, and thus provide a promising direction towards improving generative models.

## 2 BACKGROUND AND RELATED WORK

Throughout our paper, we consider the setup where we have access to a dataset $\mathcal{D} = \{x_i\}_{i=1}^n$, generated i.i.d. from some distribution $\mathbb{P}^*$ in a high dimensional ambient space $\mathcal{X} = \mathbb{R}^D$.

**Pushforward DGMs**   As mentioned in the introduction, we leverage DGMs – in particular pushforward DGMs – in order to verify the disconnectedness of the support of $\mathbb{P}^*$. We call a *pushforward model* any DGM whose samples $X$ are given by:

$$Z \sim \mathbb{P}_Z \quad \text{and} \quad X = G(Z), \tag{1}$$

where $\mathbb{P}_Z$ is a (potentially trainable) base distribution in some latent space $\mathcal{Z}$, and $G : \mathcal{Z} \rightarrow \mathcal{X}$ is a neural network. We highlight many popular DGMs fall into this category, including (Gaussian) variational autoencoders (VAEs) (Kingma & Welling, 2014; Rezende et al., 2014), normalizing flows (NFs) (Dinh et al., 2017; Kingma & Dhariwal, 2018; Behrmann et al., 2019; Chen et al., 2019; Durkan et al., 2019), generative adversarial networks (GANs) (Goodfellow et al., 2014), and Wasserstein autoencoders (WAEs) (Tolstikhin et al., 2018).

**Intrinsic dimension estimation**   If we assume that $\mathbb{P}^*$ is supported on the closure of a $d$-dimensional manifold embedded in $\mathcal{X}$ for some unknown $d < D$, a natural question is how to estimate $d$ from $\mathcal{D}$.[2] We follow Pope et al. (2021) and use the Levina & Bickel (2004) estimator with the MacKay & Ghahramani (2005) extension, given by:

$$\hat{d}_k := \left( \frac{1}{n(k-1)} \sum_{i=1}^n \sum_{j=1}^{k-1} \log \frac{T_k(x_i)}{T_j(x_i)} \right)^{-1}, \tag{2}$$

where $T_j(x)$ is the Euclidean distance from $x$ to its $j^{\text{th}}$-nearest neighbour in $\mathcal{D} \setminus \{x\}$, and $k$ is a hyperparameter specifying the maximum number of nearest neighbours to consider. While other estimators have been recently proposed (Block et al., 2021; Lim et al., 2021; Tempczyk et al., 2022), we stick with (2) throughout this work as it is well-established in the literature. As we will see, popular image datasets exhibit different intrinsic dimensions in separate regions of data space.

---

[1] The disjoint union of $d$-dimensional manifolds is a $d$-dimensional manifold (Lee, 2013): the possibility of having different intrinsic dimensions separates the union of manifolds hypothesis from the manifold hypothesis.

[2] Requiring the support to be the closure of a manifold (or a union thereof) rather than the manifold itself is merely a technicality due to the formal definition of support, which is always a closed set. See Appendix A.

**On the relevance of understanding low-dimensional structure**    As mentioned in the introduction, the manifold hypothesis provides intuition as to why deep learning has been so successful. One might naïvely believe its usefulness is merely conceptual, yet uncovering and exploiting the structure underlying data of interest is a highly relevant problem (Bronstein et al., 2021) which has recently started receiving increased attention. Pope et al. (2021) use (2) to estimate the intrinsic dimension of image datasets, not only finding strong evidence of low-dimensional structure, but also uncovering that datasets of higher intrinsic dimension are harder to classify. Similarly, Ansuini et al. (2019) link the intrinsic dimension of the representations of deep classifiers to their classification accuracy. Furthermore, the low-dimensional structure of data not only affects supervised methods: for example, training a high-dimensional density to model data lying on a low-dimensional manifold through maximum-likelihood can result in manifold overfitting (Dai & Wipf, 2019; Loaiza-Ganem et al., 2022), a phenomenon resulting in $\mathbb{P}^*$ not being learned, even if an infinite amount of data was observed. Indeed, there is a rapidly growing body of work aiming to account for the manifold hypothesis within DGMs (Gemici et al., 2016; Rezende et al., 2020; Brehmer & Cranmer, 2020; Mathieu & Nickel, 2020; Arbel et al., 2021; Kothari et al., 2021; Caterini et al., 2021; Ross & Cresswell, 2021; Cunningham et al., 2022; De Bortoli et al., 2022; Ross et al., 2022). All these works highlight the relevance of properly understanding and accounting for the low-dimensional structure of data, an understanding we improve upon in this work.

## 3    THE UNION OF MANIFOLDS HYPOTHESIS

We will assume that the support of $\mathbb{P}^*$, $supp(\mathbb{P}^*)$, can be written as:

$$supp(\mathbb{P}^*) = \bigsqcup_{\ell=1}^{L} cl(\mathcal{M}_\ell) \subset \mathcal{X}, \tag{3}$$

where $\sqcup$ denotes disjoint union, each $\mathcal{M}_\ell$ is a connected manifold of dimension $d^{(\ell)}$, and $cl(\cdot)$ denotes closure in $\mathcal{X}$.[2] We refer to this assumption the *union of manifolds hypothesis*. Note that we make the assumption that each $\mathcal{M}_\ell$ is connected merely for notational simplicity, as we could collapse each union of manifolds of matching dimension into a single disconnected manifold, but this notation allows us to reason about different $cl(\mathcal{M}_\ell)$ as connected components of interest. Note also that the standard manifold hypothesis is equivalent to adding the assumption that all $d^{(\ell)}$s are identical. In the rest of this section we empirically verify this hypothesis on various commonly-used image datasets in two parts: first by establishing the disconnectedness of $supp(\mathbb{P}^*)$, and then by estimating the intrinsic dimension on each of its (approximate) connected components.

### 3.1    VERIFYING THE HYPOTHESIS: DISCONNECTEDNESS

Before empirically verifying that the support of commonly-used image datasets is disconnected, let us develop intuition as to why this might be the case. For example, on MNIST we can likely continuously transform any 2 into any other 2 without leaving the manifold of 2s, whereas attempting to similarly transform a 2 into an 8 would likely result in intermediate images that are neither 2s nor 8s. We conjecture that class labels, which are often available for our considered datasets, provide a sensible proxy for the connected components $cl(\mathcal{M}_\ell)$ of $supp(\mathbb{P}^*)$, precisely because one can conceive of continuously transforming any one member of any class into any other member of the same class without ever leaving the class.

In order to verify this conjecture, we leverage a shortcoming of pushforward models, namely their inability to model disconnected supports, which we formalize in the following proposition. We use measure-theoretic notation, where the model from (1) is given by the pushforward measure $G_\#\mathbb{P}_Z$.

**Proposition 1**: Let $\mathcal{Z}$ and $\mathcal{X}$ be topological spaces and $G : \mathcal{Z} \to \mathcal{X}$ be continuous. Considering $\mathcal{Z}$ and $\mathcal{X}$ as measurable spaces with their respective Borel $\sigma$-algebras, let $\mathbb{P}_Z$ be a probability measure on $\mathcal{Z}$ such that $supp(\mathbb{P}_Z)$ is connected and $\mathbb{P}_Z(supp(\mathbb{P}_Z)) = 1$.[3] Then $supp(G_\#\mathbb{P}_Z)$ is connected.

**Proof sketch**: From the formal definition of support, $supp(G_\#\mathbb{P}_Z)$ need not be equal to $G(supp(\mathbb{P}_Z))$, so the proof does not immediately follow from the fact that continuous functions map

---

[3]The condition $\mathbb{P}_Z(supp(\mathbb{P}_Z)) = 1$ is highly technical and holds in all settings of practical interest, but can be violated in contrived counterexamples. See Appendix A for a discussion.

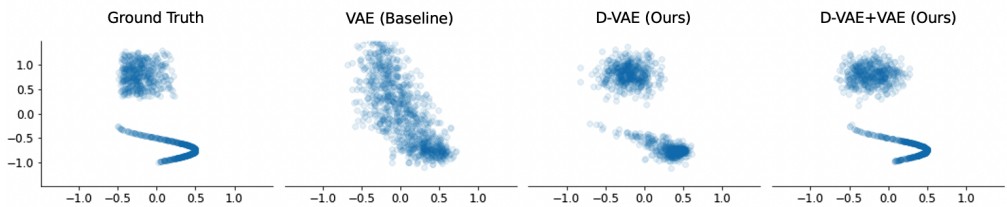

Figure 2: Samples from various VAE models on a synthetic dataset.

connected sets to connected sets. We show that $supp(G_\#\mathbb{P}_Z) = cl(G(supp(\mathbb{P}_Z)))$, and the result follows from the closure of connected sets being connected. See Appendix A for the formal proof.

The implication of Proposition 1 is simple: if the support of the data is not connected (i.e. has multiple connected components), then pushforward models are insufficient to properly model the data, since in practice $supp(\mathbb{P}_Z)$ is always connected. Intuitively, this result rules out the possibility of $G_\#\mathbb{P}_Z$ assigning 0 probability to regions of $G(supp(\mathbb{P}_Z))$ which connect different disconnected components of the data support $supp(\mathbb{P}^*)$, as illustrated in Figure 1. We highlight that topological limitations of pushforward models have been studied before (Cornish et al., 2020; Salmona et al., 2022; Ross et al., 2022), although to the best of our knowledge not to the generality of Proposition 1. Indeed, there are many methods aiming to better model disconnected supports with pushforward models (Arora et al., 2017; Banijamali et al.,

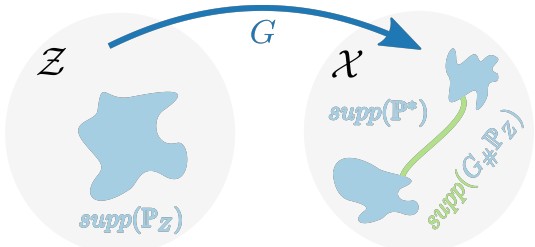

Figure 1: Depiction of the inability of pushforward models to represent multiple connected components. If $supp(\mathbb{P}^*)$ is given by the two blue connected components on $\mathcal{X}$, then by continuity, $G$ cannot map $supp(\mathbb{P}_Z)$ to this set, and so its image must contain some region (shown in green) connecting them. Proposition 1 shows that the model $G_\#\mathbb{P}_Z$ cannot learn to assign 0 probability to this green region.

2017; Locatello et al., 2018; Khayatkhoei et al., 2018; Dinh et al., 2019; Tanielian et al., 2020; Luzi et al., 2020; Pires & Figueiredo, 2020; Duan, 2021; Ye & Bors, 2021). Figure 2, which we will use as a running example throughout this section, further visualizes this result, with the first panel showing a synthetic dataset obeying the union of manifolds hypothesis (two connected components of respective intrinsic dimensions 1 and 2), along with a Gaussian VAE trained on this dataset on the second panel. The VAE fails to recover a disconnected support, further confirming this limitation of pushforward models. We include all experimental details pertaining to this section in Appendix D.1.

As a means to empirically verify that $supp(\mathbb{P}^*)$ is disconnected, we introduce *disconnected DGMs*: rather than train a single pushforward model $(\mathbb{P}_Z, G)$ on $\mathcal{D}$ as in standard DGMs, we first partition the data as $\mathcal{D} = \sqcup_{\ell=1}^L \mathcal{D}_\ell$, where $\mathcal{D}_\ell$ contains only the datapoints belonging to the $\ell^{th}$ class, and we then train a pushforward DGM $(\mathbb{P}_Z^{(\ell)}, G_\ell)$ on each cluster $\mathcal{D}_\ell$. In order to obtain a sample $X$ from a trained disconnected DGM, we simply sample $\ell$ with probability proportional to the size of $\mathcal{D}_\ell$, $|\mathcal{D}_\ell|$, and then sample from the corresponding DGM, which can be performed in a memory-efficient way (i.e. without having to load $L$ models in memory) as described in Appendix B.2.

We see disconnected DGMs as the most straightforward modification that can be done to pushforward models so as to enable them to model disconnected supports. Disconnected DGMs also allow to straightforwardly compare against standard pushforward models in a FLOP-equivalent manner that uses the same architectures (see Appendix B.1 for details). If we observe any improvement of disconnected DGMs over their standard counterparts, the **simplest explanation for this behaviour would be that** $supp(\mathbb{P}^*)$ **is disconnected**. The third panel of Figure 2 shows a disconnected VAE (denoted D-VAE), which correctly recovers two connected components (albeit not their intrinsic dimensions, which we will address later with the fourth panel), and was given the same computational budget as the VAE on the second panel: demonstrating that disconnected DGMs show improvements

Table 1: FID scores. We show means and standard errors across 3 runs.

| Model | MNIST | FMNIST | SVHN | CIFAR-10 | CIFAR-100 |
|---|---|---|---|---|---|
| WAE | $19.1 \pm 2.1$ | $51.2 \pm 2.4$ | $89.8 \pm 12.2$ | $146.7 \pm 1.2$ | $145.0 \pm 0.7$ |
| D-WAE (random) | $23.7 \pm 2.8$ | $61.1 \pm 3.6$ | $77.6 \pm 1.7$ | $154.7 \pm 0.5$ | $146.5 \pm 3.2$ |
| D-WAE (classes) | $13.5 \pm 0.2$ | $\mathbf{36.8 \pm 1.4}$ | $83.7 \pm 15.2$ | $\mathbf{133.3 \pm 0.5}$ | $209.9 \pm 3.5$ |
| WAE (GMM) | $152.7 \pm 19.1$ | $97.6 \pm 20.9$ | $276.1 \pm 8.6$ | $253.6 \pm 17.1$ | $250.3 \pm 5.0$ |
| Conditional WAE (classes) | $\mathbf{9.9 \pm 0.2}$ | $\mathbf{36.7 \pm 0.4}$ | $172.9 \pm 122.2$ | $142.9 \pm 0.5$ | $206.0 \pm 36.2$ |
| Conditional WAE (clusters) | $\mathbf{10.0 \pm 0.3}$ | $40.9 \pm 3.7$ | $\mathbf{54.1 \pm 1.6}$ | $140.7 \pm 1.1$ | $259.5 \pm 2.8$ |
| D-WAE (clusters) | $11.4 \pm 0.3$ | $\mathbf{35.3 \pm 4.3}$ | $97.6 \pm 1.4$ | $139.2 \pm 1.7$ | $\mathbf{134.3 \pm 1.8}$ |

over their standard counterparts when data is disconnected, thus providing more evidence for the validity of our test of disconnectedness.

The top half of Table 1 shows analogous results in the more realistic setting of image datasets. We use the FID score (Heusel et al., 2017) (lower is better), a commonly-used sample quality metric, to measure performance on the MNIST, FMNIST (Xiao et al., 2017), SVHN (Netzer et al., 2011), CIFAR-10, and CIFAR-100 (Krizhevsky et al., 2009) datasets. We can see that disconnected WAEs (indicated as "D-WAE (classes)") consistently outperform WAEs[4], strongly supporting our hypothesis that $supp(\mathbb{P}^*)$ is indeed not connected in these datasets. We highlight once again that disconnected WAEs were given the exact same computational budget as WAEs (both in terms of floating point operations and memory usage) to ensure fair comparisons. Nonetheless, disconnected WAEs do have more parameters. In order to ensure that the added parameters are not the reason that disconnected WAEs perform better, we perform an ablation, where we train a disconnected WAE but use random labels instead of class labels (indicated as "D-WAE (random)"). We can see that disconnected WAEs using labels are still the best performing option, once again providing strong empirical evidence to the claim that one can indeed think of classes of these datasets as approximating different connected components of $supp(\mathbb{P}^*)$. Finally, we point out that these results are not exclusive to WAEs: we carried out the same comparisons using VAEs, and found the conclusions to be the same, further backing our claims. Due to space constraints, we present these results in Table 3 in Appendix C.1.

## 3.2 VERIFYING THE HYPOTHESIS: VARYING INTRINSIC DIMENSIONS

So far we have empirically verified that commonly-used image datasets have disconnected supports and that class labels provide a good proxy for the corresponding connected components. We now confirm that these disconnected components have varying intrinsic dimensions. We achieve this by applying (2) to each $\mathcal{D}_\ell$ (the subset of the data corresponding to the $\ell^{th}$ class) separately: observing similar estimates across different values of $\ell$ would support the manifold hypothesis, whereas observing different values would instead support the union of manifolds hypothesis.

Figure 3 shows, for different values of the hyperparameter $k$, boxplots of the values $\hat{d}_k^{(\ell)}$ for class labels $\ell = 1, \dots, L$ in our considered datasets: MNIST, FMNIST, SVHN, CIFAR-10, CIFAR-100, and ImageNet (Russakovsky et al., 2015). Two relevant patterns emerge. First, within each dataset, results are mostly consistent across different choices of $k$, so that any conclusions we draw are not caused by a specific choice of this hyperparameter. Second, for all datasets except SVHN, we can observe a wide range of estimated intrinsic dimensions. These results empirically verify that the union of manifolds hypothesis is a more appropriate way to think about images than the manifold hypothesis. Additionally, we estimate the variance of these estimators through subsampling in Appendix C.2, ensuring that the observed variability of estimated intrinsic dimensions is not due to chance. We highlight that (2) is known to underestimate the true intrinsic dimension when the dataset is not large enough, which might in principle affect our estimates on CIFAR-100 and ImageNet, since these datasets have fewer datapoints per class. Strong consistency across $k$ on CIFAR-100 however suggests this is not a concern for this dataset; and even if the estimates we obtained on ImageNet do exhibit slightly more variation across $k$, the overall range of estimated intrinsic dimensions remains consistent across $k$. In other words, these results still strongly support

---

[4]Except on CIFAR-100, which we hypothesize is due to each class having too few datapoints. We provide further evidence that CIFAR-100 also has disconnected support in subsection 4.2.

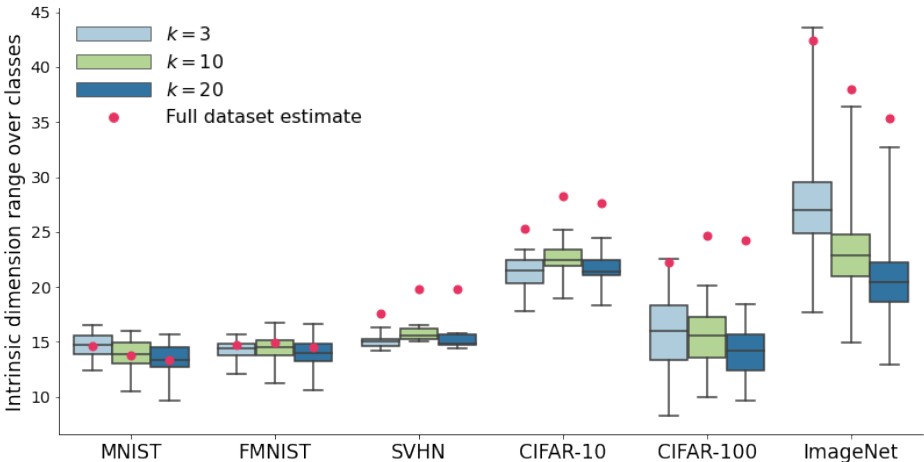

Figure 3: Boxplots showing variability of intrinsic dimension estimates across classes.

the union of manifolds hypothesis. We take a more granular look into these estimates in Appendix C.2, where we show further choices of $k$ as well as the intrinsic dimensions of individual classes.

## 4 EXPLOITING THE UNION OF MANIFOLDS HYPOTHESIS

Now that we have empirically verified the union of manifolds hypothesis on images–both the disconnectedness of the support of the data and its varying intrinsic dimensions–we turn our attention to the benefits brought to deep learning by being aware of the union of manifolds structure present in observed data. We see the results in this section as a sensible and promising first step towards realizing these benefits, and hope to encourage the community to further explore them.

### 4.1 SUPERVISED LEARNING: CLASSIFICATION ACCURACY AND INTRINSIC DIMENSION

Pope et al. (2021) showed that *datasets* of high intrinsic dimension are harder to classify. Here we provide a more granular look, by showing that *classes* of higher intrinsic dimension are harder to classify. We train 3 classifiers on CIFAR-100 (see Appendix D.2 for details): a VGG-19 (Simonyan & Zisserman, 2015), a ResNet-18, and a ResNet-34 (He et al., 2016). We focused on CIFAR-100 here as the other datasets considered in this work were either too simple to classify (MNIST, FMNIST, SVHN, CIFAR-10), or produced less-reliable intrinsic dimension estimates (ImageNet). For each class, we compute its classification accuracy and plot this against its estimated intrinsic dimension in Figure 4. We can see that, consistently across classifiers, there is an inverse relationship between estimated intrinsic dimension and classification accuracy. We also quantitatively compare intrinsic dimension and classification accuracy by computing their correlation, along with a $p$-value for independence with a $t$-test: we find that the negative correlation is significant (see figure caption). In other words, the higher the intrinsic dimension of a class, the harder it is to classify. While clearly intrinsic dimension does not fully explain accuracy, this correlation suggests the following intuition: learning useful representations for classes with higher intrinsic dimension requires learning more factors of variation and is thus more challenging.

To check if this insight can help improve classifiers, we train the ResNet-18 in two different ways (see Appendix D.2 for experimental setup): in the first, we use the standard cross entropy loss, and in the second, we weight the terms corresponding to each class in the cross entropy loss proportionally to their intrinsic dimension. This reweighting of the loss focuses more on classes of higher

Table 2: Means and standard errors of ResNet-18 accuracy on CIFAR-100 across 5 runs.

| Weights | Test accuracy |
|---|---|
| Standard | $61.38\% \pm 0.17\%$ |
| Proportional to intrinsic dimension | $\mathbf{61.77\% \pm 0.20\%}$ |

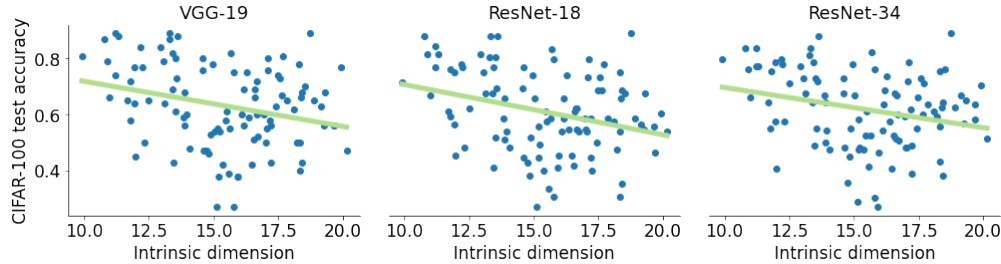

Figure 4: Class intrinsic dimension versus test accuracy on CIFAR-100, along with a least-squares regression line. Correlation coefficients for each model are $-0.243 \pm 0.015$, $-0.269 \pm 0.012$, and $-0.274 \pm 0.007$, respectively (means and standard errors over 5 runs). Corresponding $p$-values, $0.008 \pm 0.002$, $0.019 \pm 0.008$, and $0.006 \pm 0.001$, show the relationship is significant.

intrinsic dimension, as we have just shown these are harder to classify. Results are shown in Table 2, where we can see that this very simple change to the cross entropy loss marginally (but significantly, as error bars do not overlap) improves the accuracy of the network, providing an essentially "free" improvement, given the low computational overhead of estimating intrinsic dimension. Note that we did not perform any data augmentation scheme so as to not affect the intrinsic dimension of the data.

## 4.2 UNSUPERVISED LEARNING: DISCONNECTED DGMS THROUGH CLUSTERING

In subsection 3.1 we used disconnected DGMs to show that the support of image datasets is disconnected. Here we show that despite their simplicity, disconnected DGMs are performant models which can provide improvements over competing alternatives. We highlight that the aim of these experiments is not to achieve state-of-the-art performance, but rather to show that disconnected DGMs outperform comparable non-disconnected models, emphasizing the relevance of properly accounting for the union of manifolds structure present in data. To this end, we introduce a fully-unsupervised version of disconnected DGMs, where we run a clustering algorithm to partition the dataset instead of using class labels. The idea behind this modification is simply that if class labels are unavailable, a clustering algorithm might manage to recover connected components of the data support as clusters, at least approximately. Our training and clustering procedures are detailed in Appendix B.1.

We highlight once again that we use the same computational budget for training disconnected DGMs as for their standard counterparts. The only computational overhead is running the clustering algorithm itself – and more disk space is required to store the model – but the number of floating point operations and memory usage throughout training is identical (see Appendix B.1). All experimental details relevant to this section are included in Appendix D.1.

### 4.2.1 BETTER MODELLING OF DISCONNECTED SUPPORTS

As we already showed in subsection 3.1, D-WAEs outperform standard WAEs. The comparisons we have discussed so far were however meant to confirm the union of manifolds hypothesis, and remain unfair as comparisons of generative performance since the D-WAE (classes) model has access to class labels. The bottom half of Table 3 includes additional comparisons to show that D-WAEs are not just useful as a way of confirming the disconnectedness of the data support, but that they are also an effective way of accounting for this structure. We include comparisons against standard conditional WAEs, which also have access to class labels (indicated as "Conditional WAE (classes)"), but as additional inputs to their neural networks (Sohn et al., 2015), rather than having separate models for each class like D-WAEs. We can see that conditional WAEs (classes) outperform our D-WAEs (classes) only on MNIST, highlighting that D-WAEs are performant DGMs, despite their simplicity. Importantly, while we believe that D-DGMs are better than conditional models for verifying the disconnectedness of the data support (as they allow for FLOP and memory-equivalent comparisons), conditional WAEs outperforming WAEs provides further evidence supporting the union of manifolds hypothesis. Note that conditional WAEs outperform WAEs on CIFAR-100, which also supports our previous conjecture that the poor performance of D-WAEs on this dataset is due to having too few datapoints per class, rather than the union of manifolds hypothesis not holding.

We also show results for the fully-unsupervised version of the D-WAE (indicated as "D-WAE (clusters)"), and we can see that it not only outperforms the standard WAE, but that it remains competitive with the same conditional WAE, now conditioned on cluster membership rather than class labels for a fully-unsupervised baseline (indicated as "Conditional WAE (clusters)"). Finally, we also include a comparison against a WAE whose prior $\mathbb{P}_Z$ is given by a mixture of $L$ Gaussians (indicated as "WAE (GMM)") instead of a standard Gaussian as in the other models. Having a multimodal $\mathbb{P}_Z$ is a popular strategy in pushforward models to better model multimodal target distributions (Nalisnick et al., 2016; Dilokthanakul et al., 2017; Jiang et al., 2017; Ben-Yosef & Weinshall, 2018; Izmailov et al., 2020; Cao et al., 2020; Śmieja et al., 2020; Potapczynski et al., 2020). We can see that our D-WAE models outperform WAE (GMM) in all cases: this is not too surprising in light of Proposition 1, since the support of $\mathbb{P}_Z$ remains connected when using mixtures, even if it is now multimodal. Once again, we omit comparisons with other pushforward models here due to space constraints, but highlight that Appendix C.1 includes analogous comparisons with VAEs in Table 3, where we see the exact same behaviour of disconnected DGMs outperforming or remaining competitive with naïve conditioning and using mixtures as base distributions.

### 4.2.2 BETTER MODELLING OF VARYING INTRINSIC DIMENSIONS

While so far we have only focused on the ability of D-DGMs to model disconnected supports, our method also easily allows modelling varying intrinsic dimensions, thus fully accounting for the union of manifolds hypothesis. In order to do this, we simply set different dimensions for the latent spaces $\mathcal{Z}_\ell$ of each of the $L$ DGMs. In particular, we use (2) to obtain an estimate $\hat{d}_k^{(\ell)}$ of the intrinsic dimension of $\mathcal{M}_\ell$ using $\mathcal{D}_\ell$, and then set $\mathcal{Z}_\ell = \mathbb{R}^{\hat{d}_k^{(\ell)}}$. Any observed improvement of disconnected DGMs with varying $\hat{d}_k^{(\ell)}$s over disconnected DGMs with fixed $\hat{d}_k^{(\ell)}$s can thus be attributed to having accounted for varying intrinsic dimensions. For each of the $L$ DGMs $(\mathbb{P}_Z^{(\ell)}, G_\ell)$, we use two-step models (Dai & Wipf, 2019; Loaiza-Ganem et al., 2022), which were specifically designed to properly model manifold-supported data by avoiding manifold overfitting: we first train an autoencoder-type model as a first step, whose decoder is $G_\ell$; and then train a DGM $\mathbb{P}_Z^{(\ell)}$ on the low-dimensional representations obtained by running $\mathcal{D}_\ell$ through the encoder. In other words, we use a disconnected version of two-step models as a way to account for varying intrinsic dimensions. The fourth panel of Figure 2 illustrates the benefits of this approach by training a disconnected two-step VAE (indicated as "D-VAE+VAE"). This model is trained by clustering the data to obtain its connected components, estimating the respective intrinsic dimensions as 2 and 1, and then training a VAE+VAE model on each of these clusters. In the first cluster (of intrinsic dimension 2), the first VAE obtains 2-dimensional representations, and the second VAE learns the distribution of these representations. The same is done for the second cluster, except the first VAE obtains 1-dimensional representations.

In Table 4 in Appendix C.1, we carry out an analogous comparison on image datasets using VAE+NF models. Surprisingly, we found that while both versions of disconnected DGMs (i.e. with fixed or varying intrinsic dimensions) outperformed non-disconnected two-step models – once again confirming the conclusions about better modelling of disconnected supports – setting intrinsic dimensions differently on every cluster did not improve performance over keeping intrinsic dimension constant across clusters. At a first glance this might appear to imply that there is no benefit to accounting for varying intrinsic dimensions, seemingly contradicting the theoretical result of Loaiza-Ganem et al. (2022, Theorem 1), which states that manifold overfitting will occur whenever intrinsic dimension is overspecified. However, upon closer inspection of the proof of Theorem 1 in Loaiza-Ganem et al. (2022), we can see that the rate at which manifold overfitting happens depends on the difference between ambient and intrinsic dimension.[5] This observation explains both the good empirical results of Loaiza-Ganem et al. (2022) and our aforementioned results: the former are driven by the difference between ambient and intrinsic dimension (on the order of hundreds/thousands), while the latter by the error between the true and estimated intrinsic dimension (on the order of ones/tens). While a partially negative result, we provide further insight into the connection between the low-dimensional structure of the data and DGMs.

In order to verify that properly setting varying intrinsic dimensions in DGMs is advantageous, provided the true intrinsic dimensions are different enough, we generate a synthetic dataset using a pretrained BigGAN (Brock et al., 2019).

---

[5]For positive $\sigma \to 0$, Loaiza-Ganem et al. (2022) show that the likelihood of a $D$-dimensional model can go to infinity at a rate of $\sigma^{d-D}$, while not recovering $\mathbb{P}^*$, if the data is supported on a $d$-dimensional manifold.

We use this model, which has $\mathcal{Z} = \mathbb{R}^{120}$, to generate $64{,}000$ samples from the golden retriever class. These samples have true intrinsic dimension at most $120$, and their estimated intrinsic dimension through (2) is $22$ (we rounded the real-valued estimate up). We also generate another $64{,}000$ samples from this model, except we zero out all but $m = 1$ latent coordinates before passing them through the generator, resulting in samples of true intrinsic dimension of at most $m$, and estimated intrinsic dimension $3$. We then combine these two types of samples into a single dataset of size $128{,}000$, with each type of sample corresponding to a class. This dataset has the property that

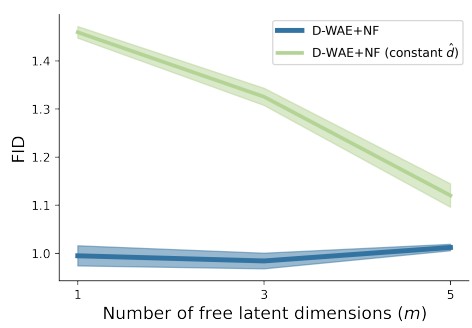

Figure 5: Mean FID scores for 3 runs on golden retriever data, the shaded areas show standard errors.

its classes have a large difference in intrinsic dimension, and estimating the intrinsic dimension of the entire dataset while ignoring classes gives a value of $5$. We then train two models on this dataset: a disconnected two-step model, where the first models are WAEs with fixed intrinsic dimension set to $5$ (i.e. the single estimate of intrinsic dimension over the entire dataset) and the second models NFs (indicated as "D-WAE+NF (constant $\hat{d}$)"); and a disconnected version of the same model except clusters have their latent dimensions set to $22$ and $3$ (i.e. class intrinsic dimension estimates), respectively (indicated as "D-WAE+NF"). We then repeat this entire process for $m = 3$ and $m = 5$, and show the resulting FID scores in Figure 5 for 3 runs. We can see that setting the latent dimension of each cluster to its estimated intrinsic dimension results in a large performance improvement versus setting it without accounting for varying intrinsic dimensions. We can also see that this gap in performance tightens as the difference in true intrinsic dimensions decreases. These results show that considering varying intrinsic dimensions is practically relevant in DGMs, particularly when the difference in true intrinsic dimensions is large.

## 5 CONCLUSIONS, LIMITATIONS, AND FUTURE WORK

In this work, we empirically verified the union of manifolds hypothesis by showing that image datasets have multiple connected components whose intrinsic dimensions vary widely. We also establish uses of the union of manifolds hypothesis, showing that classes of higher intrinsic dimension are harder to classify and how this insight can be used to improve classification accuracy. We anticipate that the broader deep learning community will further unlock the potential of the union of manifolds hypothesis for both understanding natural data and discovering empirical improvements.

Although we confirmed the union of manifolds hypothesis on commonly-used datasets, we have not tested on anything besides image data; verification of the union of manifolds hypothesis and the efficacy of disconnected DGMs on other types of data remain to be proven. We have reason to believe our improvements will generalize though, as has often been the case with other DGM innovations developed on image-like data.

We also highlight that, while our experiments show that classes are a good approximation for connected components, they do not imply an exact match: our results are consistent with some classes overlapping, e.g. 1s and 7s on MNIST might belong to the same connected component. Additionally, while we improve upon the manifold hypothesis, the realistic scenario of intrinsic dimension varying *within* a connected component is not covered by the union of manifolds hypothesis, and we believe that further probing data for this structure to be an interesting direction for future work.

Another point to note is that, while we did obtain classification accuracy improvements through the union of manifolds hypothesis, these were marginal. Finally, even though our main goal with disconnected DGMs was to verify the union of manifolds hypothesis, they can likely be improved upon as DGMs in several ways, for example through parameter sharing, or by learning the number of clusters (Salvador & Chan, 2004; Kulis & Jordan, 2012; Villecroze et al., 2022).

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

## A  DEFINITION OF SUPPORT AND PROOF OF PROPOSITION 1

Before giving the standard definition of support, we motivate it. Intuitively, the support of a distribution $\mathbb{P}$ is "the smallest set" to which $\mathbb{P}$ assigns probability 1. The idea of "the smallest set" satisfying a property is often formalized by taking the intersection of all sets satisfying the property. Naïve application of this principle however results in a "definition" of support with undesirable properties. Consider for example the case where $\mathbb{P}$ is a standard Gaussian distribution on $\mathbb{R}$. Clearly any definition of support that we use should obey that the support of $\mathbb{P}$ is $\mathbb{R}$. However, if we simply follow the "definition" of support as the intersection of all sets of probability 1, we would not obtain this: for any $x \in \mathbb{R}$, clearly $\mathbb{P}(\mathbb{R} \setminus \{x\}) = 1$, yet $\bigcap_{x \in \mathbb{R}} \mathbb{R} \setminus \{x\} = \emptyset$, where $\emptyset$ denotes the empty set. In other words, arbitrary intersections of sets of probability 1 need not have probability 1, and in fact can have probability 0. More abstractly, sets of probability 1 are not closed under arbitrary intersections, which is desirable when defining "the smallest set" satisfying a given property as the intersection of all such sets. In order to solve this problem, the support is defined as the intersection of all closed sets having probability 1 (closed sets are indeed closed under intersections).

**Definition 1 (Support of probability distributions)**: Let $\mathcal{Z}$ be a topological space. Equipping $\mathcal{Z}$ with its Borel $\sigma$-algebra to make it a measurable space, let $\mathbb{P}$ be a probability distribution on $\mathcal{Z}$. Let $\mathcal{C}(\mathbb{P})$ be the collection of all closed sets $C$ in $\mathcal{Z}$ such that $\mathbb{P}(C) = 1$. The support of $\mathbb{P}$, denoted as $supp(\mathbb{P})$, is defined as:

$$supp(\mathbb{P}) = \bigcap_{C \in \mathcal{C}(\mathbb{P})} C. \tag{4}$$

Before continuing, we point out that there exist counterexamples showing that the support of a distribution need not have probability 1 (Schilling & Kühn, 2021, Examples 6.2 and 6.3). In other words, even though closed sets are closed under intersections, closed sets of probability 1 need not share this property. These examples are however much more contrived than the one above and of no practical interest (if $\mathcal{Z}$ is a separable metric space, as is the case of $\mathbb{R}^d$, then any Borel measure $\mathbb{P}$ on $\mathcal{Z}$ satisfies $\mathbb{P}(supp(\mathbb{P})) = 1$, see Bogachev (2007, Proposition 7.2.9)), and thus support is commonly defined as above. Note that the assumption that $\mathbb{P}_Z(supp(\mathbb{P}_Z)) = 1$ in Proposition 1 could be replaced by the requirement that $\mathcal{Z}$ be a separable metric space and $\mathbb{P}_Z$ a Borel measure on it (which covers all settings of practical interest), although this would yield a slightly less general result. We restate Proposition 1 below for convenience:

**Proposition 1**: Let $\mathcal{Z}$ and $\mathcal{X}$ be topological spaces and $G : \mathcal{Z} \to \mathcal{X}$ be continuous. Considering $\mathcal{Z}$ and $\mathcal{X}$ as measurable spaces with their respective Borel $\sigma$-algebras, let $\mathbb{P}_Z$ be a probability measure on $\mathcal{Z}$ such that $supp(\mathbb{P}_Z)$ is connected and $\mathbb{P}_Z(supp(\mathbb{P}_Z)) = 1$. Then $supp(G_\#\mathbb{P}_Z)$ is connected.

**Proof**: As mentioned in the proof sketch in the main manuscript, showing that $supp(G_\#\mathbb{P}_Z) = cl(G(supp(\mathbb{P}_Z)))$, where $cl(\cdot)$ denotes closure in $\mathcal{X}$, is enough, since by continuity of $G$, $G(supp(\mathbb{P}_Z))$ is connected, and the closure of connected sets is itself connected.

We begin by verifying that $cl(G(supp(\mathbb{P}_Z))) \subseteq supp(G_\#\mathbb{P}_Z)$. Let $C \in \mathcal{C}(G_\#\mathbb{P}_Z)$. By definition of $\mathcal{C}(G_\#\mathbb{P}_Z)$, $C$ is closed in $\mathcal{X}$ and is such that $G_\#\mathbb{P}_Z(C) = 1$. By definition of pushforward measure, it follows that:

$$\mathbb{P}_Z(G^{-1}(C)) = G_\#\mathbb{P}_Z(C) = 1, \tag{5}$$

and by continuity of $G$, it also follows that $G^{-1}(C)$ is closed in $\mathcal{Z}$. Then, $G^{-1}(C) \in \mathcal{C}(\mathbb{P}_Z)$, and by definition of support, $supp(\mathbb{P}_Z) \subseteq G^{-1}(C)$. It then follows that $G(supp(\mathbb{P}_Z)) \subseteq G(G^{-1}(C)) \subseteq C$. Since this holds for every $C \in \mathcal{C}(G_\#\mathbb{P}_Z)$, we have that:

$$G(supp(\mathbb{P}_Z)) \subseteq \bigcap_{C \in \mathcal{C}(G_\#\mathbb{P}_Z)} C = supp(G_\#\mathbb{P}_Z). \tag{6}$$

Applying $cl(\cdot)$ on both sides, and using the fact that $supp(G_\#\mathbb{P}_Z)$ is closed in $\mathcal{X}$ (since it is an intersection of closed sets), yields $cl(G(supp(\mathbb{P}_Z))) \subseteq supp(G_\#\mathbb{P}_Z)$.

It now only remains to show that $supp(G_\#\mathbb{P}_Z) \subseteq cl(G(supp(\mathbb{P}_Z)))$. By definition of pushforward measure, we have that:

$$G_\#\mathbb{P}_Z(G(supp(\mathbb{P}_Z))) = \mathbb{P}_Z(G^{-1}(G(supp(\mathbb{P}_Z)))) \geq \mathbb{P}_Z(supp(\mathbb{P}_Z)) = 1, \tag{7}$$

where the inequality follows from $supp(\mathbb{P}_Z) \subseteq G^{-1}(G(supp(\mathbb{P}_Z)))$. Since $G(supp(\mathbb{P}_Z)) \subseteq cl(G(supp(\mathbb{P}_Z)))$, then $G_\#\mathbb{P}_Z(cl(G(supp(\mathbb{P}_Z)))) = 1$. It follows that $cl(G(supp(\mathbb{P}_Z))) \in \mathcal{C}(G_\#\mathbb{P}_Z)$, and thus $supp(G_\#\mathbb{P}_Z) \subseteq cl(G(supp(\mathbb{P}_Z)))$ by definition of support.

$\square$

# B   DISCONNECTED DGMS: DETAILS

## B.1   TRAINING DISCONNECTED DGMS

---

**Algorithm 1:** Training of disconnected DGMs

    **Input:** `clustering_algorithm(·)`, $\mathcal{D}$
    **Output:** $\{(\mathbb{P}_Z^{(\ell)}, G_\ell)\}_{\ell=1}^L$
1   $\mathcal{D}_1, \ldots, \mathcal{D}_L \leftarrow$ `clustering_algorithm`$(\mathcal{D})$
2   **for** $\ell = 1$ **to** $L$ **do**
3      Potentially initialize $\mathbb{P}_Z^{(\ell)}$ and $G_\ell$
4      Train $G_\ell$ and potentially $\mathbb{P}_Z^{(\ell)}$ on $\mathcal{D}_\ell$
5   **end**

---

**Training**    We summarize the training procedure for disconnected DGMs that we mentioned in the main manuscript in Algorithm 1. Note that we use the exact same computational budget to train a disconnected DGM $\{(\mathbb{P}_Z^{(\ell)}, G_\ell)\}_{\ell=1}^L$ as we do to train an equivalent non-disconnected baseline $(\mathbb{P}_Z, G)$. We begin with the observation that due to the serial nature of Algorithm 1, all $L$ models need not be in memory when training a disconnected DGM: once the $\ell^{th}$ model has been trained, it can be moved to disk before loading the $(\ell + 1)^{th}$ model to memory. In other words, even though disconnected DGMs do have more parameters than their non-disconnected counterparts and thus require more disk space to be stored, it is not needed at any point during training to load more than a single model in memory. We also make the key observation that, if using the same architecture throughout (i.e. $G$ and every $G_\ell$ have the same architecture, as do $\mathbb{P}_Z$ and $\mathbb{P}_Z^{(\ell)}$ if they are trainable), then training the non-disconnected DGM $(\mathbb{P}_Z, G)$ for $N$ epochs on $\mathcal{D}$ requires the same amount of compute as training each $(\mathbb{P}_Z^{(\ell)}, G_\ell)$ for $N$ epochs, since every $(\mathbb{P}_Z^{(\ell)}, G_\ell)$ is trained on a smaller dataset $\mathcal{D}_\ell$. To see this in more detail, assume training $(\mathbb{P}_Z, G)$ for a single epoch requires $T$ gradient steps, so the total training cost is $TN$ gradient steps. In order to train a single $(\mathbb{P}_Z^{(\ell)}, G_\ell)$ model for a single epoch, since $\mathcal{D}_\ell$ is $|\mathcal{D}|/|\mathcal{D}_\ell|$ times smaller than $\mathcal{D}$, only $|\mathcal{D}_\ell|/|\mathcal{D}|T$ gradient steps are required. Thus, training a single $(\mathbb{P}_Z^{(\ell)}, G_\ell)$ model for $N$ epochs requires $|\mathcal{D}_\ell|/|\mathcal{D}|TN$ gradient steps, and it follows that training all the $(\mathbb{P}_Z^{(\ell)}, G_\ell)$ for $\ell = 1, \ldots, L$ requires

$$\sum_{\ell=1}^L \frac{|\mathcal{D}_\ell|}{|\mathcal{D}|}TN = \frac{TN}{|\mathcal{D}|}\sum_{\ell=1}^L |\mathcal{D}_\ell| = \frac{TN}{|\mathcal{D}|}|\mathcal{D}| = TN \tag{8}$$

gradient steps. Finally, since exactly the same number of steps are required to train disconnected and non-disconnected models, and these models share architectures, it follows that the FLOP and memory costs of training them are indeed equivalent under our setup.

**Clustering algorithm**    In order to generate the dataset clusters to train each component in our disconnected DGMs (with the goal of approximating the connected components of the full dataset) without labels, we perform agglomerative clustering (Rokach & Maimon, 2005) on the dataset. At the start of the algorithm, each datapoint is in its own cluster. Then, at every step, the two clusters with the smallest linkage distance are merged, decreasing the number of clusters by 1. This occurs until a pre-specified number of clusters, $L$, is reached. For all of our datasets, we set $L = 10$ which is the number of classes for the MNIST, FMNIST, SVHN, and CIFAR-10 datasets. We experimented with ranges of $L$ between 7 and 15 and did not find a large variation in performance. Automatically inferring $L$ from a dataset is the subject of future work and would make our clustering models completely unsupervised. For all of our experiments involving agglomerative clustering, we used Ward's linkage criterion (Ward Jr, 1963). The distance between two clusters with this criterion is the variance of the Euclidean distance between all datapoints in the clusters being merged. Therefore, at each step, the two clusters with the smallest variance will be combined. We experimented with using

single linkage, which defines the distance between two clusters as the minimum distance between a datapoint in both clusters, but found that this led to very unbalanced cluster sizes and had very poor performance when training our DGMs. We also tried a custom linkage metric that merges the two clusters at each step to maximize the inter-cluster intrinsic dimension estimate variance, with the intuition that building clusters with a large variance in intrinsic dimension would better model the submanifolds of the dataset. In practice, we found this method very sensitive to initialization and saw the same unbalanced cluster failure mode as the single linkage criterion.

For all of our experiments, we also tried using the `k-means++` (Arthur & Vassilvitskii, 2007) clustering algorithm. This also worked well but gave slightly worse results than agglomerative clustering with Ward's criterion across the board.

## B.2 SAMPLING FROM DISCONNECTED DGMS

---

**Algorithm 2:** Sampling of disconnected DGMs

---

**Input:** $m$, trained disconnected DGM $\{(\mathbb{P}_Z^{(\ell)}, G_\ell)\}_{\ell=1}^L$, and corresponding cluster sizes $|\mathcal{D}_1|, \ldots, |\mathcal{D}_L|$.
**Output:** $\mathcal{S}$

1   $\mathcal{S} \leftarrow \emptyset$

2   $(m_1, \ldots, m_L) \sim \text{Multinomial}\left(m, \left(\dfrac{|\mathcal{D}_1|}{\sum_{\ell'=1}^L |\mathcal{D}_{\ell'}|}, \ldots, \dfrac{|\mathcal{D}_L|}{\sum_{\ell'=1}^L |\mathcal{D}_{\ell'}|}\right)\right)$

3   **for** $\ell = 1$ **to** $L$ **do**
4      **for** $t = 1$ **to** $m_\ell$ **do**
5         $Z \sim \mathbb{P}_Z^{(\ell)}$
6         $X = G_\ell(Z)$
7         $\mathcal{S} \leftarrow \mathcal{S} \cup \{X\}$
8      **end**
9   **end**

---

As mentioned in the main manuscript, sampling from a disconnected DGM is achieved by first sampling $\ell$ with probability proportional to $|\mathcal{D}_\ell|$, and then sampling from the corresponding DGM:

$$\ell \sim p(\ell) = \frac{|\mathcal{D}_\ell|}{\sum_{\ell'=1}^L |\mathcal{D}_{\ell'}|}, \qquad Z \mid \ell \sim \mathbb{P}_Z^{(\ell)}, \quad \text{and} \quad X = G_\ell(Z). \tag{9}$$

Sampling from (9) can be made memory-efficient, and we now describe how to do so. Assume we want to generate $m$ samples from a trained disconnected DGM model $\{(\mathbb{P}_Z^{(\ell)}, G_\ell)\}_{\ell=1}^L$. The idea is to make sure that each of the $L$ DGMs are only loaded into memory once and one at a time, which requires first knowing exactly how many samples will be required from each model. Note that, if we denote the number of samples coming from the $\ell^{\text{th}}$ model as $m_\ell$, then clearly $(m_1, \ldots, m_L)$ follows a multinomial distribution with parameters $m$ and $(|\mathcal{D}_1|, \ldots, |\mathcal{D}_L|)/\sum_{\ell'=1}^L |\mathcal{D}_{\ell'}|$. We can thus first sample from this multinomial, and then sample the appropriate number of times from each model, as described in Algorithm 2. Note that, similarly to Algorithm 1, the outer `for` loop in Algorithm 2 can be trivially made memory-efficient by loading the required model $(\mathbb{P}_Z^{(\ell)}, G_\ell)$ into memory and then removing it from memory once it has been sampled from $m_\ell$ times (and the inner `for` loop can be trivially parallelized).

## C   ADDITIONAL RESULTS

### C.1   DISCONNECTED DGMS

Table 3 shows analogous results to those of Table 1 for VAEs. We can see that D-VAEs (classes) consistently outperform VAEs and D-VAEs (random), except on CIFAR-100, further confirming the disconnectedness of the support of the data. Also, conditional VAEs (classes) convincingly outperform D-VAEs (classes) only on FMNIST and SVHN, and conditional VAEs (clusters) convincingly outperform D-VAEs (clusters) only on these same datasets, showing once again that D-DGMs provide a sensible way to account for disconnected supports.

Table 4 shows results of running a VAE+NF two-step model mentioned in subsubsection 4.2.2. We can see that the best performing disconnected versions still outperform the non-disconnected version. This is consistent with our previous results and highlights once again the relevance of accounting for multiple connected components. As mentioned in the main manuscript though, there is no significant improvement between fixing the latent dimension of every cluster to its estimated intrinsic dimension, and simply fixing the intrinsic dimension of every cluster to the same value (the estimated intrinsic dimension of the entire dataset).

Table 3: FID scores. We show means and standard errors across 3 runs.

| Model | MNIST | FMNIST | SVHN | CIFAR-10 | CIFAR-100 |
|---|---|---|---|---|---|
| VAE | $110.7 \pm 1.7$ | $100.1 \pm 0.6$ | $93.2 \pm 0.2$ | $213.8 \pm 1.3$ | $\mathbf{198.4 \pm 0.6}$ |
| D-VAE (random) | $155.4 \pm 1.3$ | $125.6 \pm 0.6$ | $115.6 \pm 0.7$ | $232.3 \pm 0.8$ | $220.9 \pm 0.8$ |
| D-VAE (classes) | $\mathbf{81.5 \pm 0.7}$ | $87.7 \pm 1.0$ | $86.4 \pm 2.0$ | $\mathbf{202.4 \pm 0.6}$ | $203.3 \pm 1.9$ |
| VAE (GMM) | $108.7 \pm 1.6$ | $99.2 \pm 0.3$ | $95.0 \pm 0.7$ | $213.7 \pm 1.1$ | $\mathbf{200.3 \pm 1.6}$ |
| Conditional VAE (classes) | $82.0 \pm 0.3$ | $\mathbf{84.7 \pm 0.9}$ | $\mathbf{71.9 \pm 1.4}$ | $211.6 \pm 1.5$ | $203.0 \pm 1.7$ |
| Conditional VAE (clusters) | $92.3 \pm 1.5$ | $105.2 \pm 1.0$ | $100.2 \pm 1.7$ | $218.8 \pm 0.3$ | $260.9 \pm 55.7$ |
| D-VAE (clusters) | $84.9 \pm 1.0$ | $106.9 \pm 0.1$ | $173.7 \pm 15.4$ | $220.8 \pm 0.9$ | $217.8 \pm 0.9$ |

Table 4: FID scores. We show means and standard errors across 3 runs.

| Model | MNIST | FMNIST | SVHN | CIFAR-10 | CIFAR-100 |
|---|---|---|---|---|---|
| VAE+NF | $119.7 \pm 2.6$ | $140.8 \pm 3.2$ | $112.5 \pm 0.4$ | $216.4 \pm 1.3$ | $204.0 \pm 0.7$ |
| D-VAE+NF (constant $\hat{d}$, classes) | $76.9 \pm 1.1$ | $86.6 \pm 0.9$ | $\mathbf{87.7 \pm 0.8}$ | $\mathbf{198.8 \pm 2.6}$ | $\mathbf{189.5 \pm 0.9}$ |
| D-VAE+NF (constant $\hat{d}$,clusters) | $75.1 \pm 0.2$ | $102.5 \pm 1.6$ | $196.5 \pm 12.1$ | $243.6 \pm 6.1$ | $207.2 \pm 0.8$ |
| D-VAE+NF (classes) | $74.5 \pm 0.3$ | $\mathbf{85.2 \pm 1.2}$ | $92.4 \pm 3.2$ | $\mathbf{201.4 \pm 1.6}$ | $\mathbf{188.1 \pm 2.2}$ |
| D-VAE+NF (clusters) | $\mathbf{73.0 \pm 0.9}$ | $99.5 \pm 0.8$ | $184.6 \pm 18.6$ | $229.0 \pm 3.9$ | $207.8 \pm 1.9$ |

## C.2 INTRINSIC DIMENSION ESTIMATES

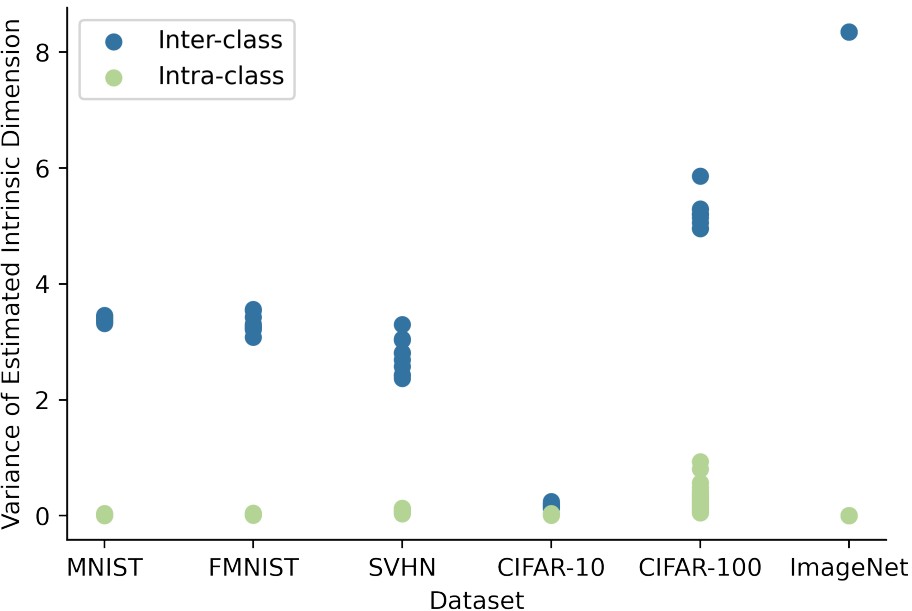

Figure 6: Variability of estimated class intrinsic dimension (with $k = 10$) across classes in blue (i.e. 11 dots, one per subsample), and across subsamples in green (i.e. one dot per class).

**Variability of intrinsic dimension estimates** In order to ensure that the variability of estimated intrinsic dimensions across classes observed in Figure 3 is caused by the true intrinsic dimensions being different rather than the variance of the estimator we used, we carry out the following experiment: For each dataset and each of their classes, we randomly subsample (without replacement) 1000 datapoints (except for CIFAR-100 where use use 300 instead, as there are only 500 datapoints per class), and estimate the corresponding class intrinsic dimension using only this reduced dataset. We repeat this process an additional 10 times, and compute the variance of estimated intrinsic dimension across classes (i.e. inter-class), and across subsamples (i.e. intra-class). Observing much lower variability across subsamples–as we do in Figure 6 for all datasets except SVHN–shows that the variability observed across classes cannot be explained by the variance of the estimator, once again supporting the union of manifolds hypothesis. Note that we sample without replacement to avoid duplicates, as these would result in a distance of 0 to their nearest neighbour.

**Intrinsic dimension of specific classes** We show a more granular breakdown of Figure 3, with intrinsic dimension estimate values for each class in Figure 7 for MNIST, Figure 8 for FMNIST, Figure 9 for SVHN, and Figure 10 for CIFAR-10. Since CIFAR-100 and ImageNet have too many classes to show, we only include the top 5 highest and lowest intrinsic dimension classes in Figure 11 for CIFAR-100, and Figure 12 for ImageNet.

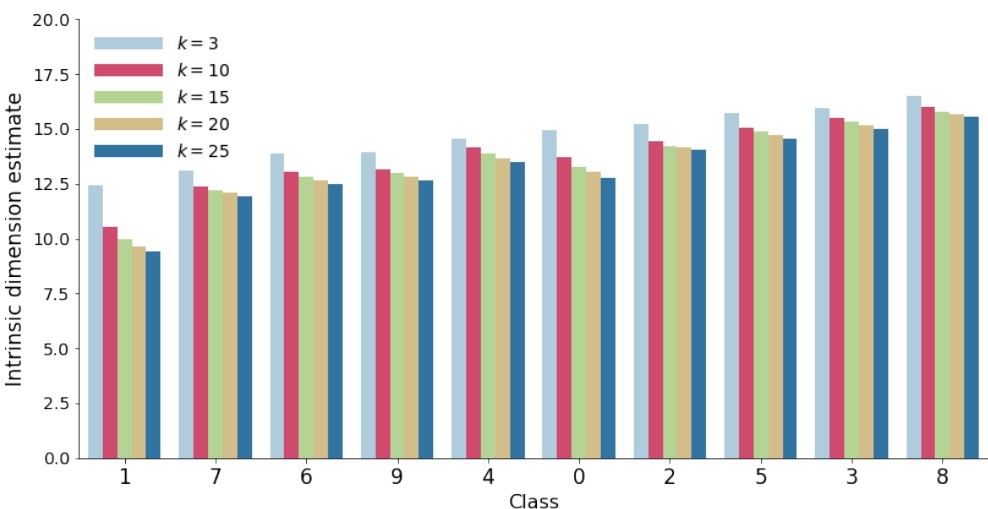

Figure 7: Intrinsic dimension estimates for classes in the MNIST dataset.

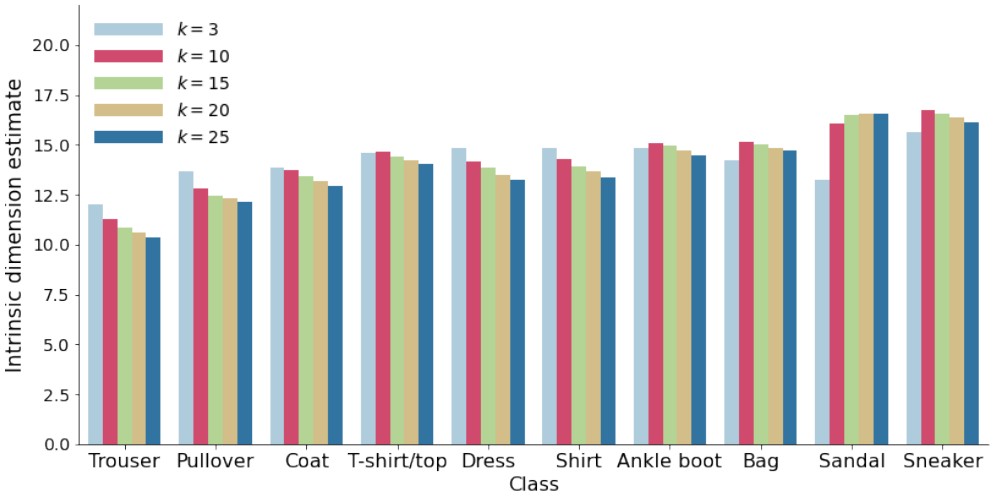

Figure 8: Intrinsic dimension estimates for classes in the FMINST dataset.

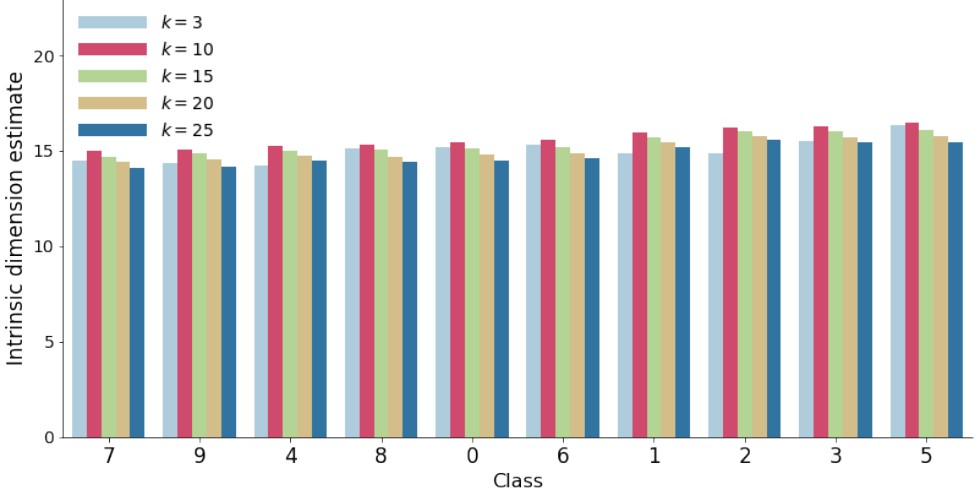

Figure 9: Intrinsic dimension estimates for classes in the SVHN dataset.

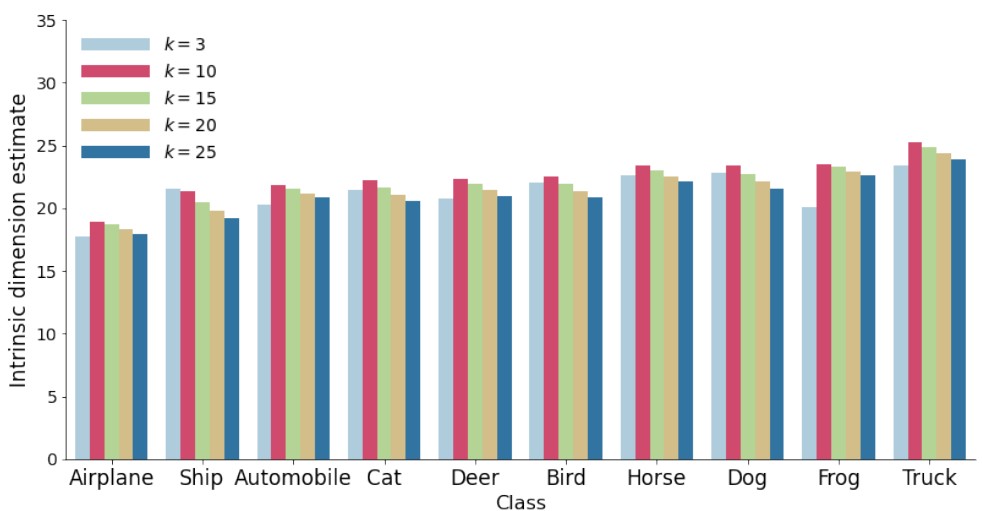

Figure 10: Intrinsic dimension estimates for classes in the CIFAR-10 dataset.

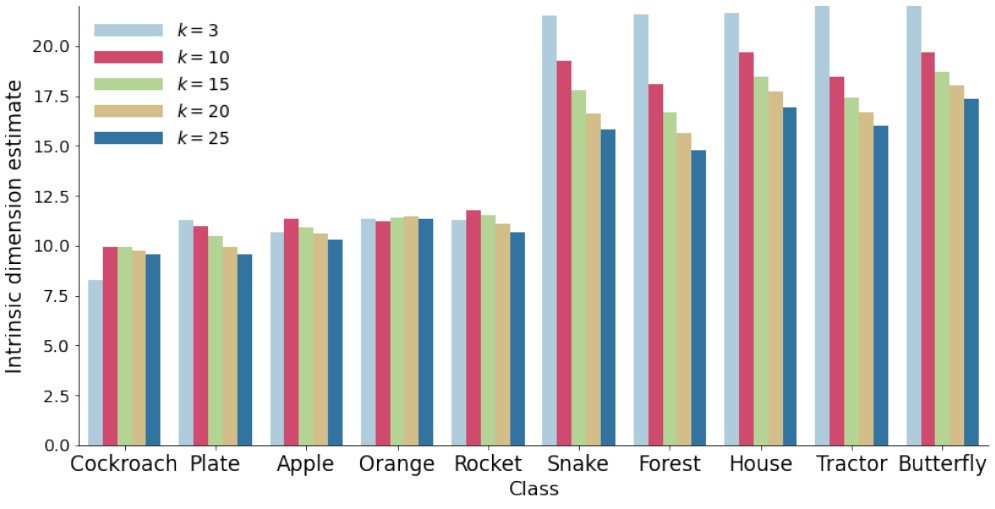

Figure 11: Intrinsic dimension estimates for classes in the CIFAR-100 dataset.

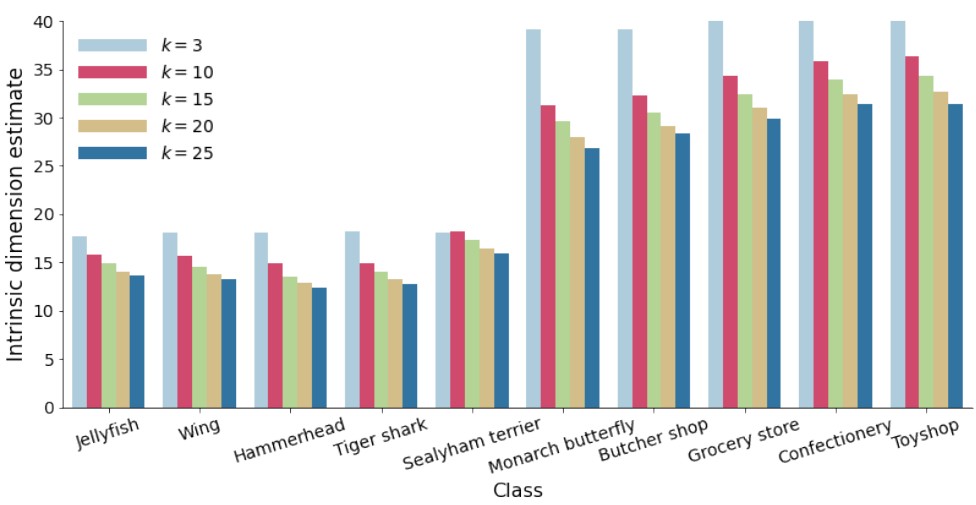

Figure 12: Intrinsic dimension estimates for classes in the ImageNet dataset.

# D EXPERIMENTAL DETAILS

## D.1 DISCONNECTED DGMS

For all models, we randomly select $10\%$ of the training dataset to be used for validation and train on the remaining $90\%$. We use a separate test dataset to report all metrics for our models. A batch size of 128 is used for all datasets. Unless otherwise noted, at the beginning of training, we scale all the data to between 0 and 1. For all experiments, we use the ADAM optimizer (Kingma & Ba, 2015), typically with learning rate 0.001 and cosine annealing for a maximum of 100 epochs. We also use gradient norm clipping with a value of 10.

**Simulated Data** To generate the ground truth data, we create a uniform mixture of two components. The first is a two dimensional square of width 1 centered at $(0, 0.8)$. The distribution along the $x$ and $y$ axis are independent beta distributions with $(\alpha, \beta)$ set to $(2, 2)$ for the $y$ axis and $(2, 5)$ for the $x$ axis. The second component is a one dimensional sinusoid (with $y$ as the input and $x$ as the output) that is one unit in length centred at 0.5 with an amplitude of 0.5 and a period of 1. The distribution along this line is a beta distribution with $(\alpha, \beta)$ set to $(2, 5)$. We generate 10,000 samples for training, 10,000 for validation and 5,000 for testing. We train three models on this simulated data. For the baseline model we use a VAE with MLPs for the encoder and decoder, with ReLU activations. The encoder and decoder each have a single hidden layer with 25 units. The latent dimension of the model is set to 2. The learning rate is set to 0.0001. We do not use early stopping and train for 200 epochs. For the second model, we train a clustering VAE with two components trained using the same setup as the baseline model. For the third model we train a disconnected VAE+VAE where each component is a two step VAE+VAE model. Both VAEs have the same architecture and are each trained the same way as the baseline model. The latent dimension of the GAE is obtained by running the MLE intrinsic dimension estimator 2 on each component.

**VAEs** For MNIST and FMNIST, we use MLPs for the encoder and decoder, with ReLU activations. The encoder and decoder have two hidden layers with 512 units. For SVHN, CIFAR-100 and CIFAR-10, we use convolutional networks. The encoder and decoder have 4 convolutional layers with $(32, 32, 16, 16)$ and $(16, 16, 32, 32)$ channels, respectively, followed by a flattening operation and a fully-connected layer. The convolutional networks also use ReLU activations, and have kernel size 3 and stride 1. The latent dimension of the model is set to 20. We use early stopping on validation loss with a patience of 30 epochs, up to a maximum of 100 epochs. For the GMM, we set the prior to have 10 modes with learnable mixture weights, means, and standard deviations. The mixture components of the prior were initialized with fixed means spaced 3 units apart centred at 0, standard deviations of 1, and uniform mixture weights. For all models, the decoder's variance is not learned.

**WAEs** For all datasets, we use an MLP with 2 hidden layers of size 256 each for the discriminator. The encoder and decoder for all datasets are the same convolutional networks used for VAEs except we increase the channels to $(64, 64, 32, 32)$ and $(32, 32, 64, 64)$ respectively. The latent dimension of the model is also set to 20. We train every model for a total of 300 epochs and perform early stopping on the reconstruction error with a patience of 30 epochs only for the GMM baselines. We note that when training for the full 300 epochs (without early stopping) the GMM baselines perform significantly worse and the other models perform better. We use a learning rate of 0.0005 for the discriminator and do not use any learning rate schedule for the model. We weight the discriminator loss with a lambda value of 10. For the GMM, we set the prior to have 10 modes with fixed means spaced 3 units apart centred at 0, fixed standard deviations of 1, and uniform mixture weights. We attempted to learn these parameters similar to the VAE but observed worse results.

**VAE+NF** For the VAE GAE, we followed the same setup as the single VAE. For the NF, we use a rational quadratic spline flow (Durkan et al., 2019) with 64 hidden units, 4 layers, and 3 blocks per layer. For all datasets, we standardize the data (ie. the latent vectors of the VAE) by subtracting the mean and dividing by the standard deviation. We do not scale the data or apply a logit transform. The latent dimension is set to the estimated intrinsic dimension of the cluster for fitted models, and set to 20 otherwise.

**WAE+NF** For the WAE, we followed the same setup as the single WAE except we use residual networks (He et al., 2016) to handle the larger and more complex golden retriever datasets. The encoder first applies a convolution (increasing the channel dimension to 16), batch normalization, and ReLU activation to the input before a max pool operation with stride 2 and kernel size of 3. The main body of the encoder is a set of 4 residual layers with channel dimensions $(16, 32, 64, 128)$. Each layer consists of two basic blocks which are two consecutive convolution, BatchNorm (Ioffe & Szegedy, 2015) and ReLU layers where the input of the block is added to the model's activations before the final ReLU. The stride of the first convolution of each layer is set to 2 to halve the spatial resolution. The skip connection is downsampled by a convolution with a stride of 2. The activations are then average-pooled across each channel and projected to the latent space dimension by a linear layer. The decoder is identical to the encoder main body except there are no downsampling operations, the spatial resolution of the activation is doubled at each layer by bilinear upsampling and the channel dimensions are $(128, 64, 32, 16)$. For the GAE, we do not perform early stopping and train for 100 epochs. The latent dimension is set to 20. The NF density estimator is also identical to the VAE+NF setup except we perform early stopping on validation loss with a patience of 30 epochs, for a maximum of 100 epochs.

### D.2 EXPLOITING THE UNION OF MANIFOLDS HYPOTHESIS: SUPERVISED LEARNING

In order to investigate if there is a correlation between intrinsic dimension and the difficulty of classification for a given class, we train three different classifiers on CIFAR-100 and measure the accuracy for each class. We use the same hyperparameters and training strategy for each of the three models we consider (VGG-19 (Simonyan & Zisserman, 2015), ResNet-18, and ResNet-34 (He et al., 2016)). We do not modify the model architecture except for changing the dimension of the last linear layer to project into a 100-dimensional vector to match the number of classes in CIFAR-100. We use the CIFAR-100 dataset test split (10,000 images) then randomly select 10% of the training dataset as our validation set (5,000 images) and use the remaining 90% as our train split. Note that this validation split is constant across all runs. The classification accuracy is reported as the accuracy on the test set on the same epoch where the best validation accuracy is achieved. We do not add any data augmentations as doing so could modify intrinsic dimension estimates and confound our results. Studying the relationship between intrinsic dimension and data augmentations is an interesting direction for future work. The combination of (1) no data augmentations, and (2) the presence of a validation set causes our results to be lower than the state-of-the-art for the models, although the results remain self-consistent here. We train each model using the cross entropy loss for a total of 200 epochs starting with a learning rate of 0.1 and decrease this to $10^{-2}, 4 \times 10^{-3}$, and $8 \times 10^{-4}$ after $60, 120$, and $160$ epochs respectively. We use the standard SGD optimizer with momentum set to 0.9 and train with a weight decay of $5 \times 10^{-4}$. For all datasets, we normalize by the channel-wise mean and standard deviation of all training images. We use a batch size of 128 for training, and a batch size of 100 for both validation and test.

We now detail the exact loss we used for our experiments with the re-weighted cross entropy loss. A weighted version of the categorical cross entropy loss is given by:

$$-\sum_{i=1}^{n} \sum_{\ell=1}^{L} \omega_\ell \cdot y_{i,\ell} \cdot \log f_\theta(x_i)_\ell, \tag{10}$$

where $y_i$ is a one-hot vector of length $L$ corresponding to the label of $x_i$, $f_\theta(x_i)$ is the $L$-dimensional output of the classifier containing assigned class probabilities, and $\omega_\ell$ is the scalar weight given to the $\ell^{th}$ class. The standard categorical cross entropy uses the weights $\omega_\ell = 1$ for $\ell = 1, \ldots, L$. Our modified weights are given by:

$$\omega_\ell = L \cdot \frac{\hat{d}_k^{(\ell)}}{\sum_{\ell'=1}^{L} \hat{d}_k^{(\ell')}}. \tag{11}$$

Note that when the intrinsic dimension estimates of all classes match, our proposed weights recover the standard ones, but more heavily weight classes of higher intrinsic dimension otherwise.

