# OpenReview forum: "Verifying the Union of Manifolds Hypothesis for Image Data"
_ICLR.cc/2023/Conference — ICLR 2023 poster_

### Official Review · Reviewer_FLmA · 2022-10-24

**Confidence:** 5
**Clarity, Quality, Novelty And Reproducibility:** The clarity is fine but the novelty i…
**Correctness:** 2
**Technical Novelty And Significance:** 2
**Empirical Novelty And Significance:** Not applicable
**Recommendation:** 5

**Strength And Weaknesses:**

Strength:
1. The union of manifolds assumption is useful in deep learning.
2. The authors provide a lot of numerical results.

Weaknesses:
1. The paper neglects a lot of important related works. First, there have been numerous works on union of subspaces and manifolds but the author failed to acknowledge them. Union of subspaces or manifolds have been studied in many works of clustering and matrix completion. See the following: [1] Vidal. Subspace clustering. 2011. [2] Elhamifar. Sparse Manifold Clustering and Embedding. 2011. [3] Fan et al. Polynomial matrix completion for missing data imputation and transductive learning. 2020.
On the other hand, deep learning based clustering methods are also closely related to union of manifolds. See the following: [4] Zhang et al. Neural Collaborative Subspace Clustering. 2019. [5] Cai et al. Efficient Deep Embedded Subspace Clustering. 2022. There are even some works proposing to train multiple autoencoders to cluster data.

2. The evidence of union of manifolds has also been observed when using t-SNE or UMAP to visualize high-dimensional data.

3. In Section 4.2, Algorithm 1 is not clearly explained. For instance, how to do the initialization and form $\mathbb{P}^{(\ell)}$?

4. The title of the paper is too broad.


**Summary Of The Paper:**

The paper aims at the union of manifolds assumption in deep learning, which states that data lies on a disjoint union of manifolds of varying intrinsic dimensions. The authors empirically verify this hypothesis on commonly-used image datasets.

**Summary Of The Review:**

My major concern is that the novelty and contribution of the paper are not significant or at least not well demonstrated. It is not clear what challenges or open problems the paper solved. Comparing with existing work on union of subspaces and manifolds, the quality of the paper is below the bar of ICLR.

---

> ### Author Response · Authors · 2022-11-11
> **Response to Reviewer FLmA (1)**
>
> We sincerely thank the reviewer for their input on our work. Overall, while we agree with several of the concerns raised by the reviewer, we see these points as being easily addressed through a manuscript update rather than fundamental flaws in the paper as suggested by the low review score, particularly given that the reviewer agrees that studying the union of manifolds hypothesis is useful. With this in mind, we kindly ask the reviewer if they would consider increasing their score, provided we address their concerns as follows:
>
> - ```The paper neglects a lot of important related works. First, there have been numerous works on union of subspaces and manifolds but the author failed to acknowledge them. Union of subspaces or manifolds have been studied in many works of clustering and matrix completion. See the following: [1] Vidal. Subspace clustering. 2011. [2] Elhamifar. Sparse Manifold Clustering and Embedding. 2011. [3] Fan et al. Polynomial matrix completion for missing data imputation and transductive learning. 2020. On the other hand, deep learning based clustering methods are also closely related to union of manifolds. See the following: [4] Zhang et al. Neural Collaborative Subspace Clustering. 2019. [5] Cai et al. Efficient Deep Embedded Subspace Clustering. 2022. There are even some works proposing to train multiple autoencoders to cluster data.```
>
> We agree with the reviewer that these works are relevant, particularly [1, 4, 5], and thank them for bringing them to our attention (note that we did cite [2]). We will cite these papers in a revised version of our manuscript. That being said, these papers are about clustering, and not about confirming the union of manifolds hypothesis (even if sometimes they are motivated by thinking the union of manifolds hypothesis holds). The relevance of our work is not from proposing clustered DGMs, but rather it is about explicitly verifying the union of manifolds hypothesis empirically. In this view, if clustering methods based on the disconnectedness of the support of the data work well, then they just provide further evidence towards the very hypothesis we are trying to prove. We are happy to add this discussion to the manuscript.
>
>
> - ```The evidence of union of manifolds has also been observed when using t-SNE or UMAP to visualize high-dimensional data.```
>
> We thank the reviewer for bringing up t-SNE and UMAP, we agree that discussing them is relevant. We respectfully disagree with the fact that these methods provide enough evidence for the union of manifolds hypothesis due to several reasons:
> 1. These methods are data visualization tools, and thus present the data through 2 or 3-dimensional representations. This makes it impossible to verify what the intrinsic dimension of different components might be whenever the intrinsic dimension is larger than 2 or 3. Furthermore, there is no guarantee that even if the intrinsic dimension of a component is 2 or 3, that it will still be represented using the same dimensionality on the t-SNE or UMAP plot.
> 2.  It is generally impossible to map high-dimensional points (even if these happen to live on a low-dimensional manifold, or a union thereof, embedded in high-dimensional space) to 2 or 3-dimensional space in such a way that all pairwise distances are preserved. It follows that the distances observed between points in these visualizations should be taken with a grain of salt. It is thus not unreasonable to believe that these methods might map two disconnected components on data space to a single connected component in 2 or 3-dimensional space in order to accommodate these topological constraints, or similarly map a single connected component on data space to disconnected components in 2 or 3-dimensional space.
>
> In summary, observing classes being clustered in a t-SNE or UMAP visualization does not by itself provide strong enough evidence supporting the union of manifolds hypothesis. This being said, observing these clusters is indeed consistent with the hypothesis as well, and we will happily include these visualizations in the updated version of our paper.

---

> > ### Author Response · Authors · 2022-11-11
> > **Response to Reviewer FLmA (2)**
> >
> > - ```In Section 4.2, Algorithm 1 is not clearly explained. For instance, how to do the initialization and form $\mathbb{P}^{(\ell)}$?```
> >
> > We thank the reviewer for pointing out a part of the manuscript where we can improve clarity, as we agree details are missing. $G_\ell$ is randomly initialized. If $\mathbb{P}^{(\ell)}$ is learnable, the neural network involved is similarly initialized. If $\mathbb{P}^{(\ell)}$ is not learnable, it is actually not initialized. We will change line 3 of the algorithm to "Initialize $G_\ell$ and potentially $\mathbb{P}^{(\ell)}$" for added clarity. The details on $\mathbb{P}^{(\ell)}$ depend on the model. For example, in the VAE case $\mathbb{P}^{(\ell)}$ is taken as a fixed standard Gaussian, and in the two-step models in section 4.2.2 it is a learnable normalizing flow. We will also make it more explicit when $\mathbb{P}^{(\ell)}$ is learnable and when it is not throughout the manuscript.
> >
> > - ```The title of the paper is too broad.```
> >
> > Could the reviewer please elaborate on what is wrong with the title? While we are unsure if we are allowed to change the title after submission, we can try to do so if the reviewer proposes an alternative. Finally, even if we do not manage to change the title, we hardly see this as a strong enough reason to reject a paper.

---

> > > ### Comment · Reviewer_FLmA · 2022-12-07
> > > **After reading the response**
> > >
> > > Thanks for the detailed response. I raise my rating, though I think the significance of the work is not high.

---

### Official Review · Reviewer_6dov · 2022-10-24

**Confidence:** 4
**Correctness:** 3
**Technical Novelty And Significance:** 4
**Empirical Novelty And Significance:** 4
**Recommendation:** 8

**Clarity, Quality, Novelty And Reproducibility:**

The message of this work is clear and appears to me to novel. The writing about the empirical claims can be improved. The results appear reproducible after a quick look at the submitted code.


**Strength And Weaknesses:**

[+] Interesting and timely topic. MH has been very influential but it is perhaps time to consider more realistic variants.

[+] Empirical investigation seems well-thought out and carefully performed.

[+] Relevant theory (Prop. 1) on the inability of push-forward generative models to model disconnected supports of data distributions is included.

[+] Appreciate the inclusion of Section 4.2.2, where tension between this work's empirical results and prior theory is addressed.

[-] Empirical results appear marginal (though significant). For example in Table 1, the proposed clustered-VAE (C-VAE) outperforms on MNIST, FMNIST, and SVHN, but not CIFAR-10 and CIFAR-100. Arguably CIFAR-10/100 are more important, as they are more relevant to real data. I would like to see a fuller discussion of this point.

[-] Claims about empirical results sometimes appear overstated and not fully spelled out. For instance in Section 4.2.1 the authors write "we can see that our C-VAE (classes) performs better on most cases...". It would be better to explicitly spell out which cases perform better and which do not. Same for Appendix C.1 "We can see that, modulo a few exceptions..." Which exceptions? Are they important or not?



**Summary Of The Paper:**

This work studies the "union of manifold" hypothesis, a refinement of the manifold hypothesis (MH) wherein data is assumed to lie on a disjoint union of manifolds of varying intrinsic dimensionality. To test this hypothesis an empirical study on common image datasets  is carried out. An empirical study of clustered generative architectures, chosen to test the disjoint manifold scenario, is also performed.



**Summary Of The Review:**

The manifold hypothesis is a fundamental idea in machine learning. Given the enormous influential of MH, it is natural to investigate extensions which have greater relevance to real data. This work makes a valuable contribution to the literature by discussing a disjoint union variant of MH. This paper is an interesting topic and well-argued, although the strength of the evidence support appears small. Nevertheless I recommend acceptance.

---

> ### Author Response · Authors · 2022-11-11
> **Response to Reviewer 6dov**
>
> We sincerely thank the reviewer for their review, and appreciate their positive feedback.
>
> - ```Empirical results appear marginal (though significant). For example in Table 1, the proposed clustered-VAE (C-VAE) outperforms on MNIST, FMNIST, and SVHN, but not CIFAR-10 and CIFAR-100. Arguably CIFAR-10/100 are more important, as they are more relevant to real data. I would like to see a fuller discussion of this point. [...] Claims about empirical results sometimes appear overstated and not fully spelled out. For instance in Section 4.2.1 the authors write "we can see that our C-VAE (classes) performs better on most cases...". It would be better to explicitly spell out which cases perform better and which do not. Same for Appendix C.1 "We can see that, modulo a few exceptions..." Which exceptions? Are they important or not?```
>
> We thank the reviewer for calling this out, as we agree that a more in depth discussion of this will improve the paper. We will include the following discussion when we update our paper. For CIFAR-10, we believe the most important part is that C-VAE (classes) does better than VAE, as this is evidence supporting the union of manifolds hypothesis (particularly the disconnectedness of the support of the data). The fact that conditional VAEs (both with classes or clusters) also outperform VAE can actually be seen as further evidence supporting this claim of disconnectedness. In other words, we see conditional VAEs outperforming C-VAEs here as merely showing that C-VAEs are not the best DGMs for CIFAR-10, rather than not supporting our main claim about the union of manifolds hypothesis. As for CIFAR-100, the results are explained by the fact that there are significantly fewer datapoints per class, and so the C-VAE (classes) model will naturally struggle more, as it has an independent model for each class. On the other hand, the conditional VAE accesses class information as an additional input, and benefits from weight sharing between classes. Altogether, these results remain consistent with the union of manifolds hypothesis, the outperformance of conditional VAEs on CIFAR-100 notwithstanding.

---

> > ### Comment · Reviewer_6dov · 2022-11-17
> > **Acknowledgement**
> >
> > Thanks to the authors for addressing my concern. Your comment that for CIFAR-10, the *"important part is that C-VAE (classes) does better than VAE"* is helpful. The difference still appears small (though significant). Fewer samples per classes is a perhaps a plausible reason why "C-VAE (classes)" under-performs on CIFAR-100. But I also wonder if it is because the CIFAR-100 submanifold(s) are just not as disjoint as your hypothesis would require, and what this implies for large-scale image datasets like Imagenet.

---

> > > ### Author Response · Authors · 2022-11-17
> > > **Discussion**
> > >
> > > We thank the reviewer for their reply, and are glad we addressed their concern. While we agree that CIFAR-100 (or massive image datasets) being less disconnected could explain our results with clustered DGMs (using classes) on this dataset, we believe this is a much less plausible explanation than CIFAR-100 simply having too few classes: First, as we mentioned earlier, conditional VAEs (classes) work much better than VAEs on CIFAR-100. The conditional models do have an inductive bias to help the model account for disconnected supports (but do not have a separate model for each class), and thus these results are highly consistent with the union of manifolds hypothesis. Second, even intuitively, it is hard to imagine the support **not** being disconnected, even if we consider all natural images: if one takes the image of a cat and an image of a car, it is hard to fathom smoothly transforming one into the other while having all the intermediate images be natural images (e.g. some intermediate images would likely be semantically meaningless).

---

> > > > ### Comment · Reviewer_6dov · 2022-11-17
> > > > **Another question**
> > > >
> > > > One more additional question: suppose in a classification problem with multiple (>3) classes, most classes exist on their own disjoint manifold, but some classes actually belong to a single manifold. For instance certain subsets of MNIST's "7" and "1" classes might plausibly share a manifold. In other words, the disjoint union of manifolds hypothesis mostly holds, but not strictly. How would this affect the empirical analysis you propose here?
> > > >
> > > > Can you also comment about other cases in which disjoint union of manifolds might fail, and how realistic they might be? For instance data with intersecting manifolds would fail to be disjoint.

---

> > > > > ### Author Response · Authors · 2022-11-18
> > > > > **Discussion**
> > > > >
> > > > > We thank the reviewer for their insightful question. We agree that classes might not perfectly recover connected components, even if the overall support is disconnected, e.g. 1s and 7s might be part of the same connected component as pointed out by the reviewer; or it could also be that a single class has several connected components, e.g. the class of dogs might have several connected components corresponding to different breeds. We do not believe that this affects our conclusion that the support of the data is disconnected: if it was not, we would not have observed the same empirical results (i.e. empirical improvements of clustered DGMs over non-clustered DGMs). We simply believe that our results should be understood as "the support of the data is disconnected", and not as "each class is a separate connected component of the support of the data", even if in practice classes provide a good enough approximation to the actual connected components. We will make sure to include this discussion in our paper.
> > > > >
> > > > > As to when the union of manifolds hypothesis might break, we think this is likely domain-specific. One way in which the hypothesis could fail is, as mentioned by the reviewer, if there is overlap in the manifolds. This could happen if a single connected component has different intrinsic dimensions (e.g. a "lollipop" in $\mathbb{R}^2$ could have a 1-dimensional stick and a 2-dimensional "candy" in a single connected component). We believe this is likely realistic in many settings, and think that probing whether this structure exists in data could be very interesting future work. It could also be that intrinsic dimension is always constant (i.e. that the manifold hypothesis holds), or that there is no low-dimensional manifold structure to begin with, although we intuitively suspect that this is less realistic for most high-dimensional, highly-structured data.

---

### Official Review · Reviewer_GFby · 2022-10-27

**Confidence:** 4
**Correctness:** 2
**Technical Novelty And Significance:** 2
**Empirical Novelty And Significance:** 2
**Recommendation:** 3

**Clarity, Quality, Novelty And Reproducibility:**

The novelty of the work is pretty limited. The manuscript in its current form requires more work.

**Strength And Weaknesses:**

Here are some strengths associated with the current work :

1) The authors work towards verifying the union of manifolds hypothesis.

Here are some weaknesses associated with the current work :

1) The authors in this work make many statements without quantifying them thus treating them as obvious facts.
" ... Assuming that data lies on a single manifold implies intrinsic dimension is identical across the entire data space, and does not allow for subregions of this space to have a different number of factors of variation. ..."

The reviewer does not agree with this hypothesis from the author. Data can be lying on a single manifold with multiple basis vectors. Different classes can choose to use appropriate basis vectors and not use the rest or use them marginally. This does not imply that different sub-regions of space cannot have different factors of variation.

"... We first prove that these deep generative models (DGMs) are incapable of modeling disconnected supports. We then argue that the class labels provided in our considered datasets identify connected components (i.e. different classes are disconnected from each other), and show that training a push forward model on each class outperforms training a single such model on the entire dataset, even when using the same computational budget: this improvement is a firm indicator that the support is truly disconnected,
thus strongly supporting our thesis."

The reviewer does not agree with the author's statement above. It is much more easier to train a model once the data has been adequately clustered as such. Training a unique model for each class as compared to a single model for different classes, the hypothesis classes associated are completely different. Thus the argument is fallacious.

2) In case the classes are separable and are non-intersecting and it is possible to separate the classes adequately (as the authors discuss in Section 3.1) then it is straightforward to train models on each separate class. However this is an unrealistic expectation in the real world since training separate models can be futile unless we have a mechanism to perform perfect classification of any unseen instance which is never true. Training models for all classes allows us to use the common structure and basis to mutually help learning for different classes.

3) The authors work is specific to image datasets and not applicable in general, which the authors are kind enough to mention in their discussion. Similarly the authors claim to use agglomerative clustering in this current work. However as mentioned in the no free lunch theorem, there is no one clustering algorithm which can perfectly cluster all instances for all types of data. The authors could have made the work more useful and valuable by adding a discussion on this.

4) The authors in this work assume that the number of clusters is known which is never true in the real world setting. Also the authors' empirical results demonstrate very marginal gains. The authors do not discuss many intricate cases i.e., the case of intersecting manifolds and how to cluster them adequately.

5) The novelty/contribution of the current work is pretty limited.

**Summary Of The Paper:**

In this work, the authors verify the union of manifolds hypothesis (proposed in earlier literature) i.e, data lies on a disjoint union of manifolds of varying intrinsic dimensions. The authors provide empirical results to support the hypothesis.

**Summary Of The Review:**

The current work is pretty limited in novelty and the authors make many statements/assumptions which do not hold true in general. The work is focussed on image datasets and thus the results and conclusions might not generally applicable. The draft in its current form needs more work.

---

> ### Author Response · Authors · 2022-11-11
> **Response to Reviewer GFby (1)**
>
> We sincerely appreciate the reviewer having spent time to provide their review. We are happy they see working towards verifying the union of manifolds hypothesis as a strength of our work. However, most of their criticisms come either from having misread our paper, or from misunderstood points that we repeat throughout the paper multiple times. In particular, we believe the reviewer missed the main point of the paper, which is to verify the union of manifolds hypothesis (which we heavily emphasize in the manuscript). This is evidenced by their complaints on a perceived lack of broad applicability of the models we use to carry out this verification (clustered DGMs). We elaborate on this below, addressing the criticisms individually. If the reviewer would kindly point to the parts of the text that caused these misunderstandings, we will happily rephrase our manuscript accordingly to avoid future readers having the same issues. If the reviewer has any lingering disagreements, we would be delighted to further discuss during the rebuttal phase.
>
> - ``` " ... Assuming that data lies on a single manifold implies intrinsic dimension is identical across the entire data space, and does not allow for subregions of this space to have a different number of factors of variation. ..."
> The reviewer does not agree with this hypothesis from the author. Data can be lying on a single manifold with multiple basis vectors. Different classes can choose to use appropriate basis vectors and not use the rest or use them marginally. This does not imply that different sub-regions of space cannot have different factors of variation.```
>
> We kindly point out to the reviewer that our claim is not a hypothesis: it is a mathematical fact. Throughout our paper, whenever we use the term "manifold", we use it in the formal mathematical sense. As mentioned in the paper, the disjoint union of $d$-dimensional manifolds is itself a $d$-dimensional manifold. The intrinsic dimension is the same as the dimension $d$ of the manifold, and it is also equal to the number of factors of variation; thus our statement is accurate. We also highlight our statement is about the **number** of factors of variation, not about the factors themselves as seems to be implied by the reviewer. Please let us know if this clears up your understanding.
>
> - ``` "... We first prove that these deep generative models (DGMs) are incapable of modeling disconnected supports. We then argue that the class labels provided in our considered datasets identify connected components (i.e. different classes are disconnected from each other), and show that training a push forward model on each class outperforms training a single such model on the entire dataset, even when using the same computational budget: this improvement is a firm indicator that the support is truly disconnected, thus strongly supporting our thesis." The reviewer does not agree with the author's statement above. It is much more easier to train a model once the data has been adequately clustered as such. Training a unique model for each class as compared to a single model for different classes, the hypothesis classes associated are completely different. Thus the argument is fallacious.```
>
> The reviewer calls our argument "fallacious" because "it is much easier to train a model once the data has been adequately clustered", without properly elaborating as to why this is the case. Indeed, we agree with the reviewer's intuition that it is easier to train a model on clustered data: our argument is precisely that clustering makes the task easier **because** the support of the data is disconnected! This being said, we agree with the reviewer that the empirical experiments we perform to support the hypothesis that $supp(\mathbb{P}^*)$ is disconnected do not confirm this hypothesis with absolute certainty. Nonetheless this is true of any empirical experiment, where the best one can hope for are observations that are consistent with the hypothesis being tested. If the reviewer has any additional experiment in mind to further support our hypothesis, we will gladly carry it out during the rebuttal process.

---

> > ### Author Response · Authors · 2022-11-11
> > **Response to Reviewer GFby (2)**
> >
> > - ```In case the classes are separable and are non-intersecting and it is possible to separate the classes adequately (as the authors discuss in Section 3.1) then it is straightforward to train models on each separate class. However this is an unrealistic expectation in the real world since training separate models can be futile unless we have a mechanism to perform perfect classification of any unseen instance which is never true. Training models for all classes allows us to use the common structure and basis to mutually help learning for different classes. [...] The authors in this work assume that the number of clusters is known which is never true in the real world setting. Also the authors' empirical results demonstrate very marginal gains. The authors do not discuss many intricate cases i.e., the case of intersecting manifolds and how to cluster them adequately.```
> >
> > These comments miss the point of our paper, which is to empirically validate the union of manifolds hypothesis. The models we use to carry out this validation (or more specifically, to verify disconnectedness of the data's support), clustered DGMs, are just a tool we use, and not the main object of study of our paper. Therefore their general applicability is orthogonal to the main conclusions we draw from the use of these models. The same holds about these models requiring the number of clusters to be specified beforehand and these models not exhibiting a huge improvement over other conditional DGMs. In other words, we are not advocating to replace all DGMs with clustered DGMs, we are just empirically verifying relevant structure that is present in image data (i.e. union of manifolds): the fact that clustered DGMs turn out to be good conditional models is just the "cherry on top", as marginal as the improvements over other conditional models may be. Reviewer **6dov** agrees with us here, ``**The manifold hypothesis is a fundamental idea in machine learning. Given the enormous influential of MH, it is natural to investigate extensions which have greater relevance to real data. This work makes a valuable contribution to the literature by discussing a disjoint union variant of MH.**'', and we would like to encourage the reviewers to further discuss this.
> >
> > - ```The authors work is specific to image datasets and not applicable in general, which the authors are kind enough to mention in their discussion. Similarly the authors claim to use agglomerative clustering in this current work. However as mentioned in the no free lunch theorem, there is no one clustering algorithm which can perfectly cluster all instances for all types of data. The authors could have made the work more useful and valuable by adding a discussion on this.```
> >
> > We agree that verifying the union of manifolds hypothesis on non-image data would make our paper stronger. However, understanding the structure of natural images is a highly relevant problem in machine learning and computer vision, and we strongly believe that the lack of verifications on non-image data is not a good enough reason to argue for the rejection of our paper. For example, the work of Pope et al. [1] (which we heavily cite in our paper), where the standard manifold hypothesis is probed, has been very well-received by the community in spite of "only" considering images.
> >
> > As for the comment on our use of agglomerative clustering, this once again misses the main point of our paper, which is about verifying the union of manifolds hypothesis, and not about clustered DGMs (i.e. where we sometimes use agglomerative clustering) themselves: First, we rely on labels when verifying the disconnectedness part of the union of manifolds hypothesis, so the broad applicability of clustering methods is of no concern to our main thesis. Second, we believe that evoking the no free lunch theorem is a fallacious criticism of clustered DGMs: while indeed no clustering algorithm will work every time, this is also true of all machine learning models, and thus not a limitation of clustered DGMs in particular. If the no free lunch theorem is understood as "no method is the best under every circumstance", it becomes irrelevant, as we are only considering a subset of circumstances here, namely when the data has union of manifolds structure (which, as we argue in the paper, is a very common scenario).

---

> > > ### Author Response · Authors · 2022-11-11
> > > **Response to Reviewer GFby (3)**
> > >
> > > - ```The novelty/contribution of the current work is pretty limited.```
> > >
> > > We respectfully disagree with this statement. If the reviewer believes the work is not novel, could they please provide a citation where the goal is to empirically verify the union of manifolds hypothesis? We highlight once again that the main point of our work is the verification of this hypothesis. As such, we do not see the novelty/contribution (or lack thereof) of clustered DGMs themselves as being relevant here.
> > >
> > > Contribution-wise, we emphasize once again that understanding the structure underlying natural images is a highly relevant problem. Once again, we point out [1] as evidence of this claim, and invite the reviewer to discuss this point with reviewer **6dov**.
> > >
> > >
> > > [1] The intrinsic dimension of images and its impact on learning, Pope, Zhu, Abdelkader, Goldblum, and Goldstein, ICLR 2021

---

### Author Response · Authors · 2022-11-11
**General Response**

We thank the reviewers for the time they spent reading our paper and for the feedback they provided on our work. We are glad all reviewers see value in empirically confirming the union of manifolds hypothesis, and that our ``empirical investigation seems well-thought out and carefully performed'' (**6dov**). We reply individually to each reviewer, and will update our manuscript towards the end of the rebuttal period so as to include feedback not only from the reviews, but from the ensuing discussion as well.

---

> ### Author Response · Authors · 2022-11-16
> **Discussion reminder**
>
> We kindly remind the reviewers that the discussion period will end in 2 days. Are there any lingering concerns that we can address?

---

### Decision · Program_Chairs · 2023-01-20

**Decision:**

Accept: poster

**Justification For Why Not Higher Score:**

While I do feel that the paper deserves being accepted and do hope that machine learning community will focus more on works like this, at this time I feel that poster is the most suitable form of presentation for this paper. I think the topic is very interesting and will provoke many debates, questions, and skepticisms. All of those can be addressed directly during the poster presentation.

**Justification For Why Not Lower Score:**

Considering everything written above, I made a decision that the paper deserves being accepted.

**Metareview: Summary, Strengths And Weaknesses:**

The paper's main goal is to empirically verify the Union of Manifold Hypotheses (UMH). UMH is the relaxed version of the more standard Manifold Hypothesis (MH) and has been previously used in the literature, including in applications of clustering. UMH differs from MH essentially in two main ways: (1) UMH assumes that the support is disconnected, and (2) UMH allows different connected components of the support to have different intrinsic dimensionalities. This allows different parts of the input space to have different (number of) factors of variation. Importantly, the paper focuses on the natural image datasets.

The authors start by providing empirical evidence (Section 3.1), supporting the fact that common natural image datasets have disconnected supports (in the pixel space). They also demonstrate that the class label provides a good proxy for identifying (some of) the connected components. Then they estimate the intrinsic dimension (similarly to the way done in [1]) for separate classes and confirm that they all have different values (Section 3.2). These empirical results provide the main body of evidence supporting UMH.

The authors go beyond verifying UMH and also make several attempts (although not very impressive) of exploiting UMH in the context of deep learning in the natural image domain. In Section 4.1 they consider classification for CIFAR100 and present a simple idea, inspired by UMH, that leads to *very modest* (but statistically significant) improvements in the test accuracy of Resnet-18 model. In Section 4.2 the authors consider deep generative models and demonstrate non-trivial improvements (in terms of FID scores on smaller datasets, such as MNIST, CIFAR10, CIFAR100, etc).

All the reviewers agree that UMH is an important concept in the field of machine learning. They also agree that the paper is clearly written, exposition is very easy to follow, and that the empirical evidence is well structured and presented. The main concern of the reviewers was related to the significance of the contribution. In particular, most of the reviewers focused their attention on Section 4 and found the results presented there rather weak.

After detailed discussions (including the video conference with all of the reviewers), I decided to support the acceptance of this paper. I believe that verifying (and empirically probing into) UMH, which is a notion actively used in some of the subfields of machine learning, is a significant enough contribution. As another argument in support of this decision, I want to mention that paper [1] undertook a line of research very similar in spirit (focusing instead on MH), was published, and is gaining quite a few citations. I think the current paper is a natural continuation of the line, started in [1]. More generally, I personally believe that as a field we should focus more on studying (empirically) the properties of our datasets (not only in the natural image domain).

My positive recommendation is conditional. I ask the authors to perform a minor revision of their paper that will include the following points (apart from all of the other points discussed during the review and rebuttal phase):

(1) Make the title more specific. All the reviewers noticed (and I agree with them) that the authors focus exclusively on the domain of natural images. Even though it is quite plausible that UMH holds for many other natural domains, the only evidence presented in this paper addresses the natural images. One of the options proposed by the reviewers is "Verifying UMH for natural images”. Personally I like this option and think that it is a fair point.
(2) I will ask the authors to read their paper once again and tone down the claims, whenever necessary. For example, it has been noticed by the reviewers that UMH was already known in the field, and yet the authors say “we propose the union of manifolds hypothesis” in the introduction. Please, make sure not to overstate your contribution.

I trust the authors to make these modifications to the manuscript.

**Note From Pc:**

if the above contains the word "oral" or "spotlight" please see: "oral" presentation means -> notable-top-5% and "spotlight" means -> notable-top-25%. As stated in our emails, we are disassociating presentation type from AC recommendations

**Summary Of Ac-Reviewer Meeting:**

When going into the meeting, two out of three reviewers were voting "reject" and one reviewer was championing the paper. Importantly, the paper under review continues the line of research started in [1] and builds heavily on it.

In the meeting we established that the paper is well written, nicely executed, and does not contain any obvious flaws. The authors were very engaged in the rebuttal phase, were quick to react, and updated their preprint very swiftly, accounting for the reviewers' questions. All the reviewers agreed that the main reason behind the "reject" recommendations was the lack of novelty and significance. Therefore, for the entire meeting we focused on this exact topic and probed the question from different angles.

The paper's main contribution is to empirically verify the "union of manifold hypothesis" (UMH). UMH is the relaxed version of the manifold hypothesis (MH). Both hypotheses are standard and well known (with UMH being the lesser known one) in the literature. Paper [1]'s main contribution was to empirically verify the MH. So, in many ways, most of concerns that could be raised when considering the current paper would also apply to [1]. Also, it is important to mention that both MH and UMH deal with real world datasets (think CIFAR10, imageNet, etc) that are obviously high-dimensional and very complicated, and because of that it is impossible to *formally prove* any of them. Instead, all we can hope for is to perform series of experiments that could provide a supporting evidence in favour of these hypotheses.

During the meeting we all agreed that the relaxation provided by UMH (compared to MH) is significant and useful to the field. UMH is being used in various subfields (including clustering), where methods are developed under UMH. This establishes that UMH on its own is an important topic. Given UMH is actively used in the field, its empirical verification would be very useful. No matter how plausible a claim about real world dataset sounds, the only way to confirm or decline it is by running experiments.

My task as an AC was to judge whether this empirical confirmation grants acceptance, and I personally felt supportive. The reviewers voting for "reject" admitted that they would not fight for the rejection. This is how the meeting ended.

[1] Phil Pope, Chen Zhu, Ahmed Abdelkader, Micah Goldblum, and Tom Goldstein. The intrinsic dimension of images and its impact on learning. In International Conference on Learning Representations, 2021.

---

> ### Author Response · Authors · 2023-02-14
> **Manuscript update**
>
> Once again, we thank the reviewers and area chair for their feedback and the time they spent on our paper. As requested, we have updated the manuscript, including changing the title, toning down some claims, and making it more explicit throughout that we only consider image data.